# RAPDOR: Using Jensen-Shannon Distance for the computational analysis of complex proteomics datasets

Luisa Hemm[1,5], Dominik Rabsch[2,5], Halie Rae Ropp[1], Viktoria Reimann [1], Philip Gerth[3], Jürgen Bartel [3], Manuel Brenes-Álvarez [1], Sandra Maaß [3], Dörte Becher [3], Wolfgang R. Hess [1,6] ✉ & Rolf Backofen [2,4,6] ✉

The computational analysis of large proteomics datasets from gradient profiling or spatially resolved proteomics is often as crucial as experimental design. We present RAPDOR, a tool for intuitive analyzing and visualizing such datasets, based on the Jensen-Shannon distance and analysis of similarities between replicates, applied to the identification of RNA-binding proteins (RBPs) and spatial proteomics. First, we examine the in-gradient distribution profiles of protein complexes with or without RNase treatment (GradR) to identify RBPs in the cyanobacterium *Synechocystis* 6803. RBPs play pivotal regulatory and structural roles. Although numerous RBPs are well characterized, the complete set of RBPs remains unknown for any species. RAPDOR identifies 165 potential RBPs, including ribosomal proteins, RNA-modifying enzymes, and proteins not previously associated with RNA binding. High-ranking putative RBPs, such as ribosome hibernation factor LrtA/RaiA, phosphoglucomutase Sll0726, antitoxin Ssl2245, and preQ(1) synthase QueF predicted by RAPDOR but not the TriPepSVM algorithm, are experimentally validated, indicating the existence of uncharacterized RBP domains. These data are available online, providing a resource for RNase-sensitive protein complexes in cyanobacteria. We then show by reanalyzing existing datasets that RAPDOR effectively examines the intracellular redistribution of proteins upon growth factor stimulation. RAPDOR is a generic, non-parametric tool for analyzing highly complex datasets.

RNA-binding proteins (RBPs) are crucial components of ribonucleo-protein complexes, including ribosomes, the signal recognition particle, and CRISPR-Cas complexes and play vital roles in all domains of life. The eukaryotic RBP database lists more than 6300 ortholog groups with more than 315,000 individual RBPs across 162 eukaryotic species[1]. Several thousand RBPs have more recently been identified in mammals, including many metabolic enzymes that are also binding to RNA[2,3]. RBPs play crucial roles in various regulatory pathways and are involved in the regulation of alternative splicing[4], in neural cell maturation in mammal cells[5], cancer and epigenetic mechanisms[6], and development in plants[7].

In prokaryotes, RBPs play significant roles in the post-transcriptional regulation of gene expression. Previously described regulatory RBPs in gram-negative bacteria include Hfq[8,9], ProQ[10–12],

[1]Genetics and Experimental Bioinformatics, of Biology, University of Freiburg, Freiburg, Germany. [2]Bioinformatics Group, Department of Computer Science, University of Freiburg, Freiburg, Germany. [3]Department of Microbial Proteomics, Institute of Microbiology, University of Greifswald, Greifswald, Germany. [4]Signalling Research Centres BIOSS and CIBSS, University of Freiburg, Freiburg, Germany. [5]These authors contributed equally: Luisa Hemm, Dominik Rabsch. [6]These authors jointly supervised this work: Wolfgang R. Hess, Rolf Backofen. ✉e-mail: wolfgang.hess@biologie.uni-freiburg.de; backofen@informatik.uni-freiburg.de

CsrA[13] and KhpA/B[14]. Many RBPs have been discovered in recent years also in other groups of bacteria, such as in gram-positive *Streptococcus pneumoniae*[15]. Several RBPs were described also for different Archaea[16]. However, the complete set of RBPs has not yet been identified for any organism.

Consequently, experimental and computational methods have been developed to identify putative RBPs. In silico methods mostly rely on amino acid strings (k-mers) as features to classify the RNA-binding nature of a protein. Prominent examples for such prediction tools based on machine learning are RBPPred[17], its successor Deep-RBPPred[18], and TriPepSVM[19]. TriPepSVM was trained on k-mer frequencies of known RBPs from different organisms. However, it can also identify RBPs in cross-species predictions.

High-throughput experimental methods for identifying candidate RBPs include fractionation approaches, which are based on separating the cell lysate by density gradient ultracentrifugation or size exclusion chromatography and extracting fractions based on differences in molecular mass or buoyant density. In the Grad-Seq ultracentrifugation approach, the proteome and transcriptome composition of each fraction is determined. Overlapping protein/transcript occurrences indicate potential RBP-RNA interactions. The first application of this method identified the major RNA chaperone ProQ in *Salmonella*[20]. Grad-Seq has been used since to identify RBP candidates in different types of bacteria, including *Clostridioides difficile*, *Enterococcus* species, *Fusobacterium nucleatum*, and the cyanobacterium *Synechocystis* sp. PCC 6803 (*Synechocystis* 6803)[14,15,21–25].

To obtain a higher resolution of the captured complexome, the glycerol or sucrose ultracentrifugation gradient can be replaced with size exclusion chromatography[26]. However, the co-occurrence of a particular RNA and a particular protein is not necessarily indicating their interaction. To address this issue, two further protocols were established. R-DeeP was developed using the HeLa S3 cell line[27], while GradR was developed in *Salmonella enterica*[28]. Both methods are based on gradient fractionation of whole cell lysates, similar to Grad-Seq. But in contrast to Grad-Seq, two gradients are prepared in parallel. One gradient is loaded with cell lysate that was treated with RNase beforehand, while the other gradient serves as a control and is loaded with untreated cell lysate. After ultracentrifugation and fractionation, the protein contents of all fractions are measured using mass spectrometry. Proteins that shift in the fractions of the RNase-treated gradient were binding RNA directly (RBPs) or were part of an RNA-containing complex (RNA-dependent proteins, RDPs), as the mass of the RNA was removed and the RNP-complex disintegrated into smaller complexes upon digestion. The R-DeeP study identified 1784 RNA-dependent proteins in HeLa S3 cells. Of these, 537 proteins lacked a previous link to RNA[27]. In *Salmonella*, the RBP FopA was identified using this method[28].

Since these types of experiments produce large and complex datasets, their computational analysis is just as important as the experimental design. While analysis pipelines have been published along with these experiments, their flexibility regarding the number of fractions and replicates is often limited. For example, R-DeeP fits Gaussian models to the curve representing the mean mass spec profile from the three replicates for each condition. To ensure as many peaks as maxima are found in the profile, the location, value and standard deviation estimate for each maximum found was provided for the Gaussian model. In a second step, Gaussian models were fitted to each replicate. A Student's *t* test was then used to assess the *p*-value (FDR-corrected) for the difference between the Gaussian fits of control and treatment, indicating shifts that are associated with RNA dependencies[27]. However, the protein abundances throughout the different fractions do not necessarily follow Gaussian distributions and the R script used to analyze the experiment was fixed to 25 fractions and three replicates per group. Adjusting this script for a different number of fractions requires considerable manual effort.

In another approach, hierarchical clustering was employed to analyze GradR data and discover RBPs clustering with known ones[28]. Although in this way the FopA protein was discovered as a previously unknown member of the family of FinO/ProQ-like RBPs[28], this methodology lacks a straightforward way to account for experimental and biological variance by using replicates.

To overcome existing limitations, we developed a tool based on the Jensen-Shannon Distance (*JSD*) and the analysis of similarities (ANOSIM), called **R**apid **A**NOSIM using **P**robability **D**istance for estimati**O**n of **R**edistribution (RAPDOR). RAPDOR can be used to analyze any distribution of proteins over a fractionation analysis with two conditions (e.g., RNase treated vs. control). Since RAPDOR is independent of the fractionation approach, it can handle data resulting from ultracentrifugation as well as size exclusion chromatography.

As a direct application, we used RAPDOR for the analysis of protein complexes after gradient profiling with or without RNase treatment (GradR) in the model cyanobacterium *Synechocystis* 6803. Cyanobacteria differ from most non-photosynthetic bacteria by the presence of extensive intracellular photosynthetic membrane systems, the thylakoids. But it is the presence of these membrane systems that likely triggered a particular set of RRM domain-containing RBPs to get involved in the intracellular transport and localization of certain mRNAs[29,30]. Moreover, there is extensive evidence for the presence of post-transcriptional regulation in cyanobacteria[31–36]. However, information on regulatory RBPs and RNA chaperones in cyanobacteria is limited. Homologs of enterobacterial RNA chaperones, such as CsrA, ProQ, FinO, or Hfq, are missing in cyanobacteria or do not bind RNA[37]. For *Synechocystis* 6803, previous Grad-Seq analysis yielded a small number of potential RBPs[25]. One of these proteins, the YlxR homolog Ssr1238, was recently verified as involved in tRNA maturation[38].

Here, we provide a list of potential RBPs in *Synechocystis* 6803 identified by RAPDOR using the generated GradR data and computationally predicted by a custom version of the TriPepSVM algorithm. In addition, we apply RAPDOR to existing spatial proteomics datasets from HeLa cells[39] to showcase its broader suitability. In that dataset, the redistribution of proteins among cellular compartments under various stress conditions was investigated. While the original publication employed a tailored approach using parametric tests to identify the distribution shifts, our findings demonstrate that the generic and non-parametric RAPDOR workflow produces similar results with fewer statistical assumptions, thus being more robust and less prone to outliers in different application scenarios. RAPDOR, as a flexible-to-use tool, is available as a PyPI package with its source code and its documentation on https://domonik.github.io/RAPDOR/. The code used to analyze the data is available as a Snakemake workflow on GitHub (https://github.com/domonik/synRDPMSpec). The GradR data is available online in a static version of RADPOR at https://synecho-rapdor.biologie.uni-freiburg.de, providing a comprehensive resource for the identification of RNase-sensitive protein complexes in *Synechocystis* 6803.

## Results
### GradR analysis of an unicellular cyanobacterium
Triplicates of *Synechocystis* 6803 cultures were grown under moderate light conditions (50 μmol m$^{-2}$ s$^{-1}$) in BG11 medium, and cell lysates were prepared when an OD$_{750}$ of 0.9 was reached. The cleared cell lysates were split into two parts of identical volumes. One part of each lysate was RNase-treated, while the second was mock-treated for control (Fig. 1A). Then, all lysates were subjected to GradR ultracentrifugation, but using β-D-maltoside (DDM) as a soft membrane solubilizer and sucrose gradients instead of glycerol. Proteins that exhibited a shift in fractions were either bound to RNA directly (RBPs) or lost their association to an RNA-containing complex (RNA-dependent proteins). Gradients showed a distinct color profile after centrifugation, relating to the specific pigment-containing complexes of *Synechocystis* (Fig. 1).

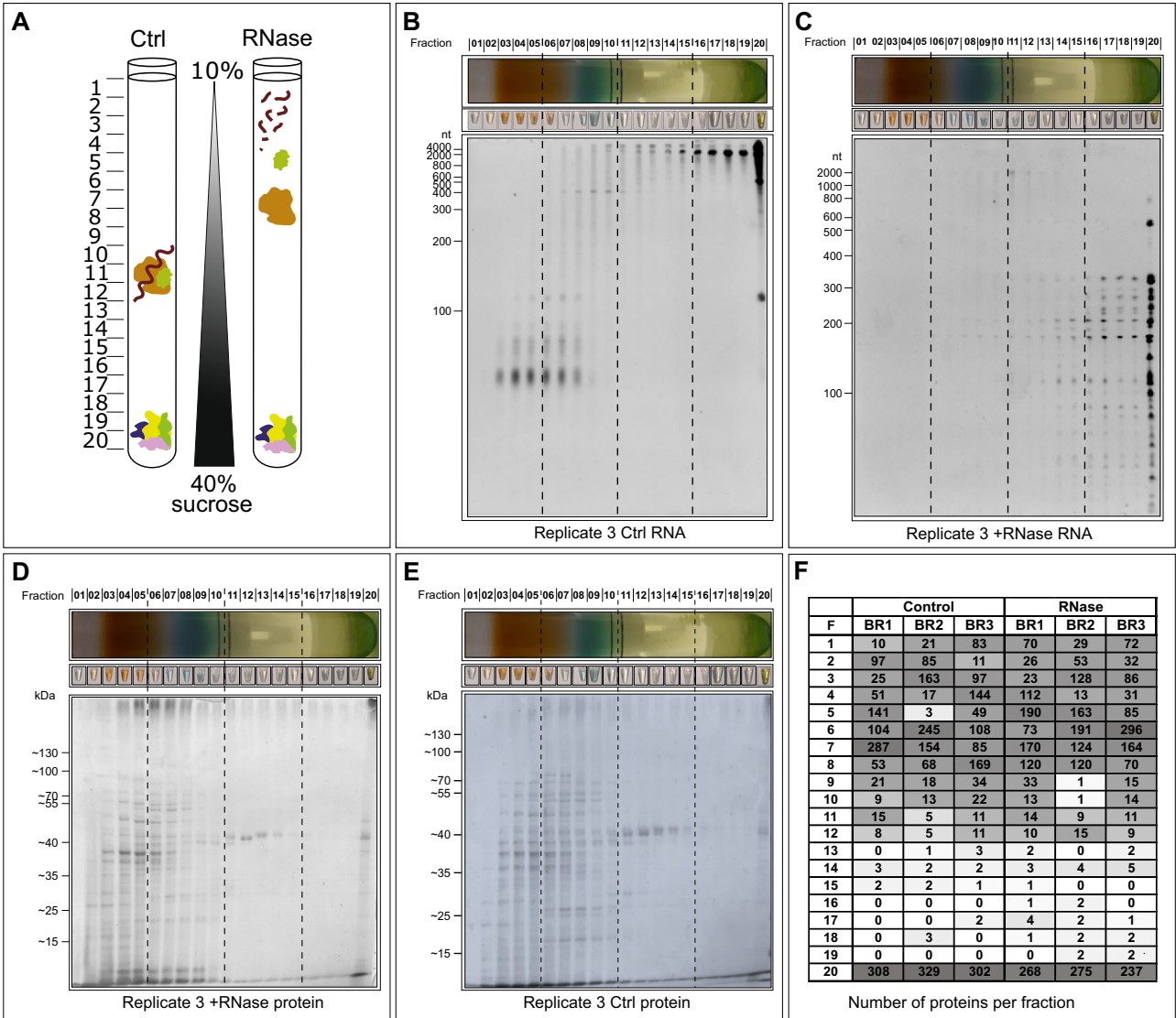

**Fig. 1 | Experimental approach and fractionation of gradients. A** Experimental setup to identify RNA-dependent proteins. Shifts in the in-gradient position of proteins following RNase digestion can result from the loss of the mass of RNA (brown wavy line) that was bound to an RBP, or the disruption of RNA-bridged protein complexes, meaning that shifting proteins had been associated with an RNA-containing complex (yellow protein). **B** Separation of the extracted RNA of the untreated control sample replicate 3 on a 10% denaturing gel stained with ethidium bromide verifies RNA quality from the different fractions. **C** RNA from the different fractions in the treated replicate 3 (after RNase treatment). **D** Protein distribution for the RNase-treated fractions. **E** Protein distribution for the fractions without RNase. In panels (**D** and **E**), a 12% SDS-PAGE was loaded with 20 µL per fraction of replicate 3 and stained with InstantBlue Coomassie Protein Stain (Abcam). **F** Overview on GradR fraction complexity and distribution. For each fraction, the number of different proteins with the highest abundance in that fraction is given. Data are shown for the three biological replicates BR1–BR3, with or without (control) RNase. The shading gradient goes from dark gray (many peaking proteins) to white (few or no proteins peaking in this fraction). Source data are provided as a Source Data file.

In the untreated sample, the RNA was distributed along the gradient with concentration peaks in fractions 3–6 for short RNAs and fractions 16–20 for longer RNAs. Most RNA was in pellet fraction 20 (Fig. 1B). As a proof of concept, the distribution of some known RNA within the fractions is shown in Supplementary Fig. 1. The RNase P RNA RnpB[40] was detected in fractions 6–13, with a peak in fractions 9 and 10. A similar distribution plus presence in fraction 20 was obtained for the transfer-messenger RNA (tmRNA) SsrA[41], while the sRNA PmgR1[32] was detected in fractions 5–12, with a peak in fractions 8 and 9 (Supplementary Fig. 1). After the addition of RNase, all RNA was digested in fractions 1–10, as well as the high molecular weight RNA in fractions 11–20. In the last fractions, some RNA remained, mainly < 400 nt in length. In fraction 20 more remaining RNA was observed, indicating a protective effect of the co-fractionating ribosomal proteins (Fig. 1C).

The protein composition of each fraction was determined by mass spectrometry analysis, measuring 100 µL aliquots of each fraction. In total, 1134 proteins were detected, 6 were found in only one replicate, 62 in two replicates, and the rest had unique peptides in all three replicates (Supplementary Data 2). Hence, ~ 31% of the annotated 3.681 proteins in *Synechocystis* 6803 were identified, which relates to the fact that only material from a single growth condition was analyzed. Nevertheless, all samples showed good correlation (Spearman correlation coefficients of Intensity-Based Absolute Quantification (IBAQ) values ≥ 0.89 between replicates from the same treatments, and ≥ 0.85 between replicates from different treatments), and the principal component analysis revealed a clear separation of RNase-treated and control samples already in the first principle component (24.32% variance explained; Supplementary Fig. 2). Besides the mass spectrometric analysis, the distribution of proteins was visualized by SDS-PAGE with

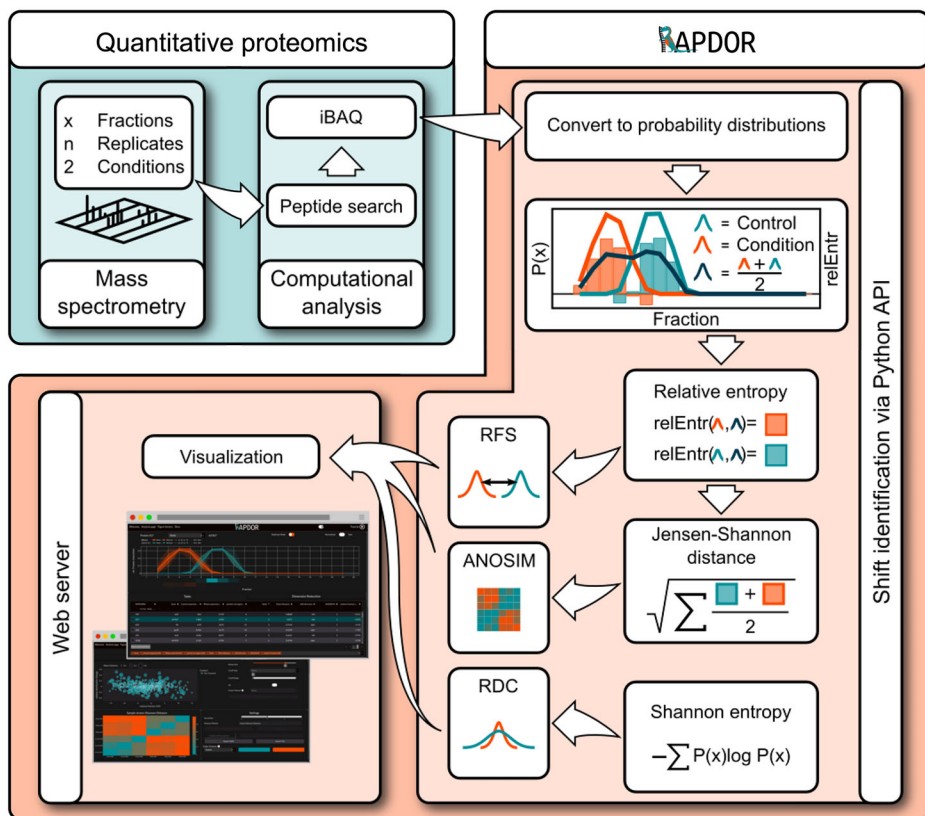

**Fig. 2 | RAPDOR workflow.** The RAPDOR tool converts analyzed mass spectrometry data with x fractions from n replicates and two conditions into probability distributions. It continues with the calculation of the relative entropy (relEntr) and the Shannon entropy. It uses the Jensen-Shannon distance (*JSD*) between all pairwise samples to carry out an ANOSIM and evaluates the effect size via the *JSD* between the mean distributions of the conditions. Lastly, distribution changes are visualized in an interactive web server displaying several parameters such as the relative position shift (RSD) and the relative distribution change (RDC). Depending on the number of replicates, the individual proteins can be either tested for redistribution using ANOSIM or ranked according to their assigned ANOSIM *R* values.

subsequent Coomassie blue staining (Fig. 1D, E). In the gel, no difference between the treated and untreated samples was observed. Most proteins were present in fractions 3-8 and in the pellet fraction 20. An overview of fraction complexity and distribution is given in Fig. 1F. Plotting the sedimentation of selected protein complexes to the respective fractions in comparison to the calculated molecular masses indicated a resolution limit of ~50 kDa (Supplementary Fig. 3). A regression analysis showed a linear relationship between the calculated molecular masses and their fraction distribution.

## A tool for the analysis of protein compartment/fraction distribution profiles

For the analysis of in-gradient distribution profiles, we developed a tool based on the *JSD*[42], called RAPDOR. The analysis workflow of such profiles consists of basically three steps: 1) preprocessing, 2) detection of significantly differential profiles between the treated and untreated samples, and 3) prediction of the profile shift between conditions (Fig. 2). Concerning preprocessing, the input for the RAPDOR workflow is a csv file containing a row for each protein and a column for each fraction, and sample, respectively. Our tool further allows the use of an averaging kernel of adjustable size to smooth the distributions along the fraction dimension. This can reduce the variance among replicates since experimental fractionation is usually not 100% reproducible. The smoothed intensities are then normalized to add up to 1. This results in a probability distribution function $P_j^i$ where $x \in X$ is the fraction number, $i$ marks the corresponding protein and $j$ the sample. In preparation for the second step (detection of significantly different profiles), it calculates the mean distribution $\widehat{P}_t^i(x) = \frac{1}{n}\sum_{j \in t} P_j^i(x)$ for the treated ($t = +$) and untreated ($t = -$) $n$ replicates for each protein.

Concerning the second and third steps, the RAPDOR workflow uses a non-parametric approach for the detection of significantly different profiles. To the best of our knowledge, the only other approach to determine profile differences and shift directions is R-DeeP[27]. R-DeeP, however, uses a parametric approach by assuming a Gaussian model fit to the curve representing the distribution profile from the three replicates for each condition. In addition, it uses the parametric Student's *t* test to evaluate the significance of peak shifts, which are found by the Gaussian fitting process. Especially, the assumption of a Gaussian mixture model for the distribution profile is critical, as there is only a small number of fractions, which leads to boundary effects that are poorly modeled by a Gaussian mixture model. Consequently, we find that R-DeeP has problems in analyzing proteins of large complexes, such as the ribosomal RNA-binding protein RpsQ, where the untreated condition usually has a large value in the last fraction, which is then distributed to nearby lower fractions in the treated condition (Supplementary Fig. 4).

For our non-parametric approach, we compare the profiles between treated and untreated samples by interpreting them as probability distributions $P_t^i(x)$ (*i* being the protein, *t* being the condition). A natural selection to determine the effect size is then a metric between probability distributions. Thus, we use the *JSD* as default to evaluate the effect size via the $JSD(\widehat{P}_+^i(x) || \widehat{P}_-^i(x))$ (Eq. 1)) of the mean distributions. Internally, the *JSD* uses the KL divergence (Eq. 2) of two distributions to their mixture distribution, resulting in a divergence measurement that is a metric and thus symmetric. Under the null hypothesis that the treated and untreated distributions are equal, the two distributions will be the same or at least very close, resulting in a small average KL-Divergence to the mean distribution. On the other

hand, an increasing KL-Divergence to the mixture distribution gathers information against this null hypothesis. In addition, the *JSD* is constrained within the range of 0 to 1 when using a logarithmic base of 2. A value of 0 shows that the two distributions do not overlap at all, while a value of 1 indicates that they are identical, thus enhancing the interpretability of the default effect size measure. Also, it is not negatively influenced by the high number of zero values measured for most proteins along the gradient in GradR. Thus, the *JSD* (i.e., our definition of effect size) between mean profiles per condition is part of the ranking provided by RAPDOR.

While the effect size, as defined by the *JSD* is valuable information for step 2 (detection of significantly differential profiles), it does not provide a rating of significance. For statistical significance, RAPDOR makes use of the replicate structure of the GradR experiments or other fractionation protocols, which typically have three replicates. Using *JSD* on all pairs to measure the effect size for a specific pair of replicates for the same protein, we generate a matrix of differences (or dissimilarity) for each protein. A popular non-parametric test on a matrix of dissimilarity was introduced with the ANOSIM *R* statistics (Analysis of similarities[43]). Here, the matrix of dissimilarity corresponds to a set of samples, each belonging to a single site. The idea of the ANOSIM *R* statistics is simply to evaluate the rank similarities within the same condition/site and between conditions/sites. The null hypothesis $H_0$ is that the similarities between conditions are greater or equal to the similarities within a condition. The corresponding test statistics *R* is the difference between the average rank similarities of pairs of samples from the two different conditions ($\overline{r_B}$) and the average rank similarities of pairs of samples from the same condition ($\overline{r_W}$, see Eq. 7) and "Methods").

As mentioned previously[43], the *R* statistic itself is a useful comparative measure of the degree of separation of sites (in our case, conditions). This implies that we can use the calculated *R*, for each protein to rank proteins according to their likelihood of binding RNA, providing a more sensitive ranking than the effect size on mean profiles. That allows, for example, a user of the RAPDOR workflow (Fig. 2) to look at the *R* statistics of some known RNA-binding proteins, and to investigate all proteins with an *R* higher than the known RNA-binding proteins.

Depending on the number of replicates, it is even possible to calculate a *p*-value. For this purpose, we implement the permutation test for $H_0$ as introduced in the original ANOSIM paper (see also "Method" for details). The basic idea is to generate all possible permutations of test/untreated labels for the replicates of a specific protein, and then calculate the *R* value, which provides an experimental distribution for the *R* values for this protein. However, for the typical number of $n = 3$ replicates and $k = 2$ conditions, there are only $(2*3)!/((3!)^2*2!) = 10$ (ref. 43, Eq. 8) possible distinct permutations, one being the original matrix. Thus, we cannot generate significant results with three replicates using the permutations for each protein individually. For that reason, RAPDOR allows to generate permutations for all proteins and conditions to determine a sampling distribution. For *i* proteins, this results in $(n*2)!*i$ many *R* values from distinct permutations for *a* dataset (Supplementary Fig. 5). Using the distribution of these value, RAPDOR enables the calculation of a *p*-value for each protein. However, when using a small number of replicates as typically used in these fields (e.g., three to four replicates), the statistical power will still be low after a correction for multiple testing and control of the false discovery rate. However, users can counteract this via either adjusting their significance level or using the *R* value ranking instead of *p*-values.

In the third step, we need an automatic way to evaluate the direction and the length of the shift. Here, RAPDOR computes, for each protein, the contribution of each position to the KL-divergence between the mean distributions $\widehat{P}^i_{+/-}$ to the corresponding mixture distribution (Eq.)). Peaks can then be defined as the positions with maximal KL divergence contribution. However, plateau-like peak profiles would result in noisy peak detection. For that reason, we apply a temperature-scaled soft-argmax function to determine the peak locations. This procedure highlights the expected position of the strongest shift $S_t$ in the treated and untreated replicates. Via subtracting the two positions, the shift length is determined as a value called the relative fraction shift. A shift is called left if the subtraction results in a negative value and right if it is greater than zero. Noteworthy, this calculates a shift direction also for very similar distributions. Therefore, it is crucial to differentiate shifts based on the mean distance of their peaks and ANOSIM, consequently interpreting the direction of the shift only when it is clearly discernible.

Thus, one interesting information is whether, for a given protein, the mean distribution does have a similar Shannon entropy in both conditions or not. A similar entropy would e.g., occur when we have a clear peak that is only shifted in its location in the two conditions. The entropy would be different, though, if there is a clear peak in one condition, which is flattened in the other condition. Thus, the developed tool further evaluates whether a shift led to a broader or sharper distribution. Subtracting those two entropies yields a single number, which we call the relative distribution change. Hereby, positive values mean that the protein had a much broader distribution after treatment. In contrast, a low negative value indicates that the protein accumulated in a single fraction and was before uniformly distributed along the gradient. In combination, the bubble plot displays the entropy difference (*y*-axis), the fraction shift and direction of the two calculated peaks (*x*-axis) together with the effect size (size of the bubble), yielding an excellent tool for a fast selection of candidate proteins. The overall workflow is described in Fig. 2.

## RAPDOR is well-suited to analyze bacterial GradR data

For the *Synechocystis* 6803 GradR data, the ribosomal protein RplA was used as a control for the shifted position in gradient fractions after RNase digestion. RplA is a member of the uL1 ribosomal protein family and directly interacts with the 23S rRNA[44,45], making it a suitable example for RNase-induced shifts. In the mass spectrometry data as well as in the experimental verification by Western blot analysis, RplA showed a very clear left shift from fraction 20 to fraction ~4 (Fig. 3A). This shift was picked up and visualized well in the RAPDOR analysis (Fig. 3A). RAPDOR indicated for the small ribosomal subunit proteins a general shift to lower molecular weight fractions, whereas the large ribosomal unit proteins did not completely disintegrate upon RNase digestion, with a few proteins, such as RplA, showing strong shifts and others remaining in high molecular weight complexes (Fig. 3B). This behavior was also visible in the histograms. Most of the proteins belonging to the small subunit showed a low effect size shift that was reproducible among the replicates, as indicated by the distribution of the mean *JSDs* and the high ANOSIM *R* values. In contrast, the distribution of *R* values for proteins from the large subunit resembled the incomplete disintegration, as only some proteins showed larger values, but most were centered around 0. A similar observation was also made in another study[46]. We assume that partial ribosomal subunit complexes remain because they are also stabilized by protein-protein interactions, or because some rRNA fragments were inaccessible for the RNase treatment, consistent with some remaining RNA fragments in fraction 20 (Fig. 1C). When sorting the dataset according to a decreasing ANOSIM *R* and a decreasing *JSD*, the median rank of proteins from the small ribosomal subunit was much smaller than the median rank of proteins from the large ribosomal subunit (Fig. 3C). No shift was observed for the purely protein-based photosystem I and II complexes indicating that their in-gradient distributions were not influenced by RNA-binding (Fig. 3B). We conclude that the experiment as well as the RAPDOR algorithm worked well for identifying known RNA interaction partners.

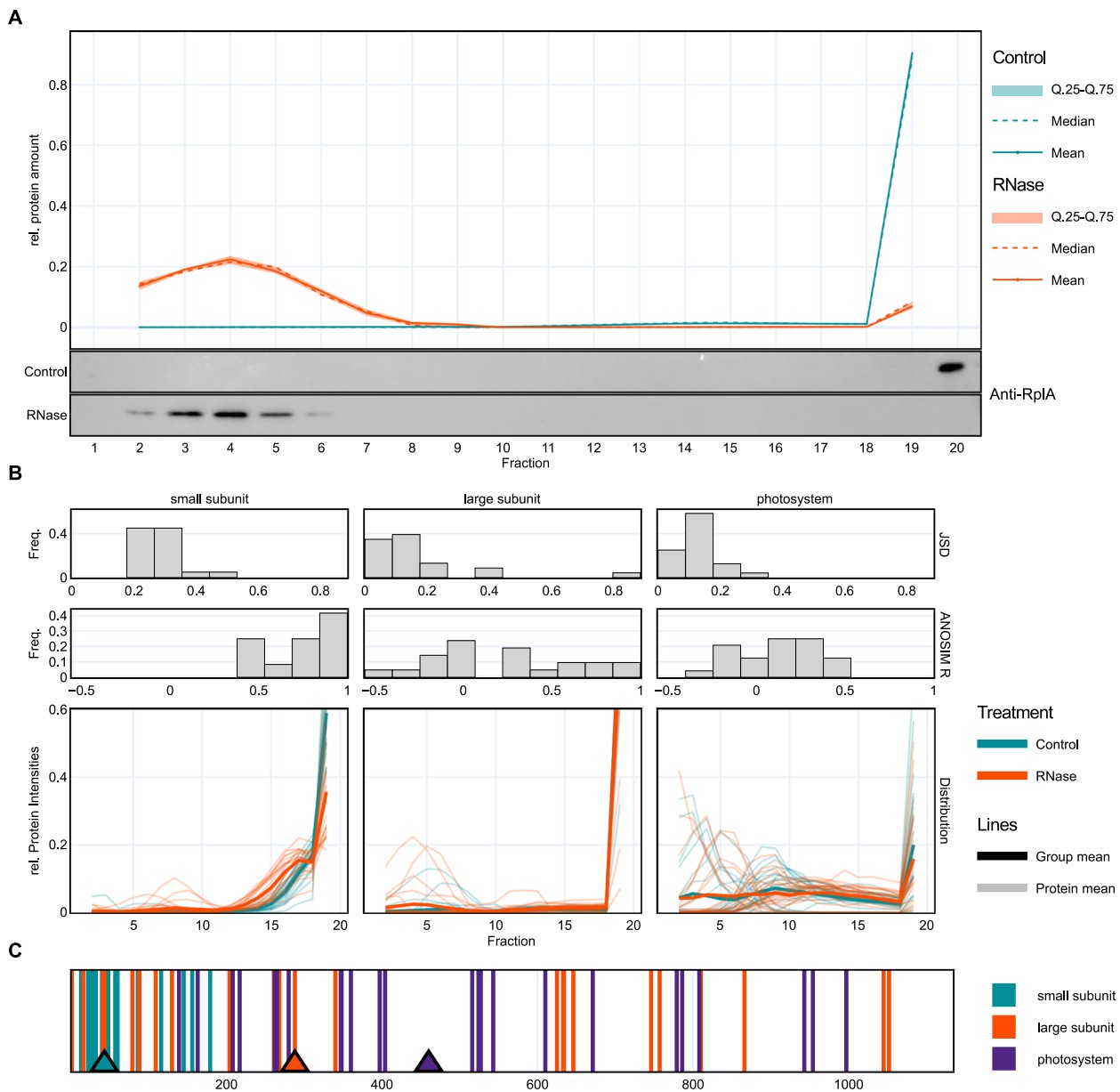

**Fig. 3 | The shift of ribosomal proteins in the RNase-treated fractions shows proof of principle. A** Shift of large subunit protein RplA was detected by western blot of replicate 1 samples and mass spectrometry data analyzed by RAPDOR (all replicates). **B** Almost all proteins of the ribosomal small subunit show a leftward shift, supported by a high ANOSIM $R$ and a $JSD > 0.2$. The large ribosomal subunit disassembles with few proteins, showing a strong shift to lower molecular weight fractions, such as RplA. Proteins of the photosystems do not shift. **C** Plot of ranked RAPDOR median values for the three protein groups from panel B (triangles). The median for small subunit ribosomal proteins is at 23, while the medians of the other two groups were at much larger ranks (smaller values indicate higher probability of RNA binding). Source data are provided as a Source Data file.

## Comparison of experimentally and SVM-predicted RBP candidates

To determine the ANOSIM cutoff, we used the 95 percentile of the distribution of ANOSIM $R'$s from all proteins generated using every possible permutation of the treatment labels (Supplementary Fig. 5). Although the number of proteins within these thresholds (ANOSIM $R$ value $\geq 0.481$, $p$-value $\leq 0.05$) was limited and their selection might result in a relatively high false discovery rate, we opted for this procedure as it is still sensitive enough to identify promising candidates for further investigation. In addition, we considered the TripPepSVM-generated list of 306 candidate RBPs with a score $\geq 0.25$ (Supplementary Data 2). For further benchmarking, we checked the number of ribosomal proteins that were detected by the different approaches.

TripPepSVM classified 45 of the 52 ribosomal proteins as RBPs (Supplementary Fig. 6). By mass spectrometry, 52 ribosomal proteins were identified. Based on the experimental evidence, RAPDOR detected 29 ribosomal proteins as RBP, while 23 were below our ANOSIM $R$ threshold of 0.481. In contrast, R-DeeP identified shifts for 4 ribosomal proteins. In the RAPDOR analysis, 11 of the 29 ribosomal proteins detected as RNA-dependent exhibited the highest achievable ANOSIM $R$ value of 1 and a $JSD$ exceeding 0.2 (Supplementary Data 2).

Besides the ribosomal proteins, 28 other proteins detected by mass spectrometry harbor the GO term for RNA binding in the *Synechocystis* 6803 Uniprot annotation (from a total of 108 annotated RBPs including ribosomal proteins). These 28 proteins were part of the SVM training data and thus classified by TriPepSVM as RBPs. RAPDOR found

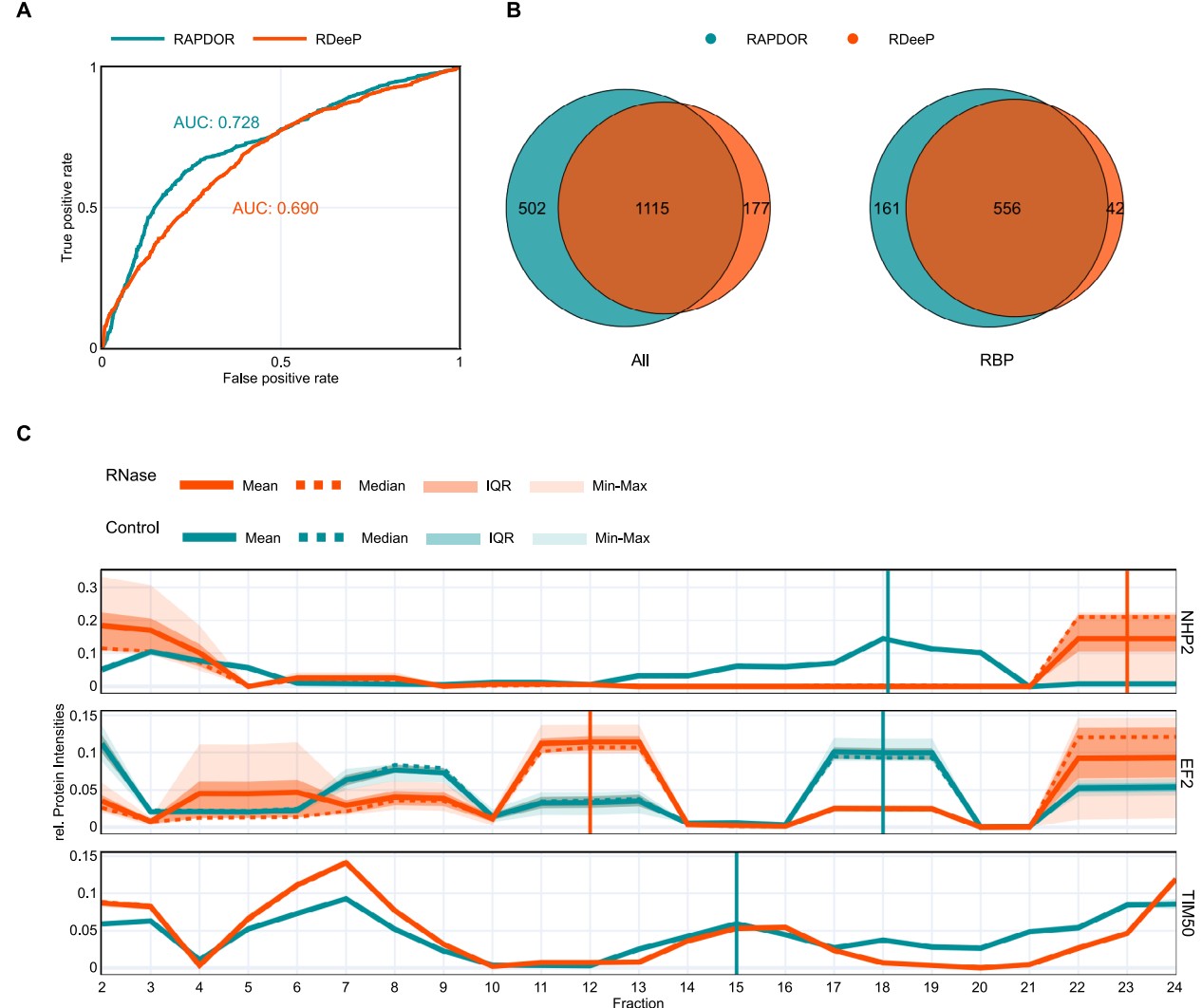

**Fig. 4 | RAPDOR vs R-Deep in human RBP classification. A** Receiver operator curve for the R-DeeP dataset ranked via RAPDORs ranking and a ranking using a decreasing *p*-value calculated via the R-DeeP tool. RAPDOR achieved a higher AUC (0.728) compared to R-DeeP (0.69). **B** Venn diagrams show the overlap of R-DeeP and RAPDOR-identified proteins. The left diagram displays all identified proteins, while the right diagram shows how many of those are known RBPs. **C** Distribution profiles of examples identified by only one of the two tools. Vertical stripes indicate the position of the *t*-tests at the Control (blue) and the RNase (orange) peak used by R-DeeP. Solid line indicates the mean; dotted line indicates the median. The darker shaded area shows the interquartile range, and the lighter shaded area shows the full range (min–max). While identification of a shift via R-DeeP highly depends on the noise at the selected peaks, with EF2 being identified as RBP and NHP2 not, RAPDOR struggles with profiles containing noise spread over the distribution, such as in EF2. R-DeeP also fails to pick fractions for testing that are not at a Gaussian peak, as in the profiles of TIM50. Ideally, the control peak test would be carried out around fraction 20. TIM50 is missing a highlighted RNase test due to a different handling of the averaging kernel between RAPDOR and R-DeeP. The position of the RNase test was, however, significant and at fraction 25. Source data are provided as a Source Data file.

six of these annotated RBPs, including the RRM-domain containing protein RbpA on rank 67, and none was detected by R-DeeP.

## Enhanced RBP classification performance with RAPDOR

Since RAPDOR is a non-parametric approach for analyzing gradient fractionation data, we further compared the performance of RAPDOR and the parametric R-DeeP pipeline using the original R-DeeP dataset[27]. In their experiment, the authors performed an experiment very similar to GradR in human HeLa cells. Normalized intensity values for each protein were used as input for either of the tools. R-DeeP assumes a Gaussian distribution for the protein profiles as the underlying model. Using the parameters of the fitted multi-Gaussians for control and treated samples, two FDR-corrected *p*-value for the significance of the differences between the groups were calculated using a Student's

*t* test, for the maximal peak (or shoulder region) in each sample group. RBPs are defined using a *p*-value cut-off 0.05 for both *p*-values (resp. the maximum of both *p*-values). In contrast, RAPDOR generated a ranking of proteins using a decreasing *R* value and mean distance (Supplementary Fig. 7). To assess the capability to identify RBPs from this kind of data, we calculated the area under the receiver operator curve (AUROC) and the area under the precision recall curve (AUPRC) of both tools, considering proteins with the GO-term for RNA binding as positives and others as negatives. To determine positive predictions for the AUC calculations, we use as a threshold parameter the maximal rank to be classified as positive in case of RAPDOR, and a varying cut-off for both *p*-values (equivalent to a *p*-value cut-off for the maximum of both *p*-values) for R-Deep. RAPDOR outperformed R-DeeP with a higher AUROC of 0.728 compared to 0.69 (Fig. 4). Especially the steep

incline at the beginning of both curves highlights that either of the tool's positions known RNA binders among its first ranks. However, RAPDOR achieved a better AUPRC, scoring 0.54, while R-DeeP scored 0.52 (Supplementary Fig. 7C). We further compared the absolute number of detected RBPs. Therefore, we took proteins with a maximum $p$-value of 0.05, corresponding to the default setting of R-DeeP, and the top candidates with an R value of 1 into consideration. The R value cutoff was determined using the 95 percentile of R values calculated from all treatment label permutations. Both tools showed a large overlap of 1115 such proteins, from which 556 were known RNA binders (Fig. 4B). While 502 proteins (including 161 known RBPs) reached the maximum $R$-value of 1 in RAPDOR but were not identified by R-DeeP, the latter detected a significant shift for 177 proteins (42 known RBPs) that did not meet the $R$-value cutoff.

To investigate the differential discovery potential of the tools, we compared profiles of known RBPs that were only detected by one of the tools (Fig. 4). Both tools are susceptible to false negatives when distribution profiles are very noisy. RAPDOR, e.g., assigned an $R$ value of 0.407 to the RBP EF2. In contrast, R-DeeP's performance highly depends on the fractions selected for the $t$-tests. To pick these fractions, R-DeeP employs an algorithm that fits a multi-Gaussian model to the mean of the data. For each treatment, control and RNase, it then selects either a peak or a shoulder region (see Caudron-Herger et al. [27] for algorithmic details), whichever shows the greatest difference compared to the other group. A $t$ test is then performed on the selected region using Gaussian fits of the individual replicates. For EF2, this algorithm selected two fractions without much noise, thus resulting in two $p$-values below 0.05. However, the peak selection procedure only uses the mean and does not account for the variance in the data. Thus, it selected the very noisy peak at fraction 23 for NHP2, which led to a non-significant test. Since other fractions were less noisy, this shift was, however, picked up by RAPDOR with an ANOSIM R of 1. Due to the parametric nature of the model, R-DeeP fails to detect changes in the distribution if they do not happen at Gaussian peaks or shoulder regions. This was, for example, the case for the RBP TIM50. The largest gain in the control samples was in a valley between two peak regions around fraction 20. R-DeeP did not capture this change, and the $t$-test was instead carried out at fraction 15. RAPDOR, in contrast, identified this shift, as the profiles showed nearly no noise, and the overall distribution clearly changed between conditions.

Noteworthy, the experimental setup of R-DeeP also aims to identify proteins that interact with RNA indirectly, e.g., via a protein-protein interaction (PPI) with an RBP. To ensure that RAPDOR's superior performance is not simply due to missing information about these proteins, we repeated the experiments with an additional adjustment. This time, we included proteins with high-confidence PPIs (score > 0.7) involving an RBP in the STRING database as part of our positive examples. Using this updated dataset showed that out of the 1617 proteins that exhibited a shift with an $R$ value of 1, 1178 (72.9%) were either directly associated with RNA or indirectly linked through a high-confidence PPI. R-DeeP also achieved a notable true positive rate of 73.1%. However, RAPDOR outperformed it in terms of AUROC and AUPRC, scoring 0.66 and 0.76, respectively, compared to R-DeeP's 0.63 and 0.74 (Supplementary Fig. 7A, C).

## Runtime comparison between RAPDOR and R-DeeP
Both tools used for analyzing experimental data demonstrated low memory consumption, with approximately 150 MB for the *Synechocystis* GradR dataset and around 300 MB for the HeLa dataset from the R-DeeP publication[27] (Supplementary Fig. 8). However, RAPDOR outperformed the R-DeeP pipeline in terms of runtime. RAPDOR processed the *Synechocystis* dataset in 2 s, whereas R-DeeP required ~ 2 min. Notably, R-DeeP struggled to scale efficiently with the increasing number of detected proteins. This issue became evident when analyzing the HeLa dataset, which contains roughly twice as

many detected proteins (3042). RAPDOR completed the analysis in 6 s, compared to around 30 min for R-DeeP. Furthermore, RAPDOR demonstrated superior performance even in more complex scenarios. It remained faster when calculating $p$-values for a simulated dataset with nine replicates per group, outperforming R-DeeP even when the latter was run with only three replicates.

## Candidates for *Synechocystis* proteins interacting with RNA from GradR data
Proteins with a high ANOSIM $R$ value, an SVM prediction as RBP, or both, represent promising candidates for further research. Therefore, following the RAPDOR analysis, the results were filtered for an ANOSIM $R \geq 0.481$, yielding 165 proteins, including 29 ribosomal and 136 non-ribosomal proteins (Supplementary Data 2). From the latter, we selected 20 proteins for more detailed analysis based on additional criteria (Table 1). These criteria were the behavior of homologs in GradR experiments in another cyanobacterium, *Nostoc* sp. PCC 7120 (*Nostoc* 7120)[46], homologies to other known RBPs or a promising SVM score. Two proteins in this list are known RBPs, namely Ffh, the apo-protein of the signal recognition particle[47] and the cold-induced RRM domain protein RbpA/Rbp1[48,49]. These two proteins were highly ranked also by their SVM score (on ranks 19 and 16, Table 1), which is likely a direct effect of training the TriPepSVM algorithm with known RBPs. With Ssl3335/SecE we found one more component of the protein translocation machinery in our short list (RAPDOR rank 72). Another protein in Table 1 supported by TriPepSVM was Sll1967, placed by RAPDOR on rank 2 and by TriPepSVM on rank 282, a putative homolog of the 23S rRNA (uracil(1939)-C(5))-methyltransferase RlmD[50]. Interestingly, we also found enzyme subunits of the chlorophyll biosynthesis pathway ChlD, ChlM and ChlI among the candidates for RNA-dependent enzymes. We included the $Mg^{2+}$ chelatase subunit ChlI (Slr1030) in Table 1 because of its TriPepSVM classification, although its ANOSIM $R$ value was slightly below the threshold. Magnesium chelatase is a heterotrimeric enzyme (subunits ChlD, ChlI and ChlH) that generates Mg-protoporphyrin IX in the first committed step in the pathway to chlorophyll[51]. Our data suggest the subunits ChlI and ChlD (RAPDOR rank 34) as RNA-dependent (while ChlH was not detectable), possibly via the catalytic subunit ChlH[52], which also functions as an anti-sigma factor[53]. With ChlM (Slr0525, $Mg^{2+}$-protoporphyrin O-methyltransferase), also the enzyme following in the chlorophyll biosynthesis pathway was classified as RNA-dependent (RAPDOR rank 74, Supplementary Data 2).

Another six proteins in Table 1 have a connection to known ribonucleoprotein complexes or into RNA metabolism. Slr1265 is the RNA polymerase subunit RpoC1[54], Sll7087 is the type III-Bv CRISPR-Cas protein Cmr4[55], RaiA/Lrt appears as a homolog of the ribosome-associated translation inhibitor A[56] (Supplementary Fig. 9), Ssl2245 is a putative antitoxin to the PemK/ribonuclease-type toxin Sll1130[57], QueF is a homolog the enzyme 7-cyano-7-deazaguanine reductase involved in biosynthesis of the modified nucleoside queuosine, a process recently characterized as relevant for bacterial lifestyle regulation[58]. Another interesting protein is Ssr3189, a small protein of unknown function conserved in cyanobacteria, its homolog Asl3888 in *Nostoc* 7120, was in a polynucleotide kinase (PNK) assay validated as RBP[46]. In addition to Ffh and Ssr3189, with Slr2130, a third protein from our short list showed an RNase-dependent shift, also for the homolog in *Nostoc* 7120[46]. Slr2130 is the enzyme 3-dehydroquinate synthase AroB, involved in the biosynthesis of aromatic amino acids. Three more enzymes qualified as RNA-dependent proteins, the vinyl reductase domain containing Slr0147, the L-amino acid dehydrogease Slr0782, and the phosphoglucomutase Sll0726. Finally, two transcription factors were included in Table 1, Sll1371, the cAMP receptor protein SyCrp1[59] and the GntR-type transcription factor Sll1961[60]. All proteins from Table 1 are marked in Fig. 5 and their distribution profiles are given in Fig. 6. Proteins that shifted to lower molecular

**Table 1 | Selected RNA-dependent candidates based on RAPDOR analysis and manual filtering**

| Protein | ANO | JSD | Shift | Rank | SVM | Comments and annotation |
|---|---|---|---|---|---|---|
| **Sll1967** | **1** | **0.85** | **− 4** | **2** | **282** | **RNA methyltransferase RlmD, Co-enriched with the RNase P-interacting protein YlrR/Ssr1238[38]** |
| *Slr1531 | 1 | 0.74 | + 12 | 6 | 19 | Signal recognition particle protein Ffh |
| Slr0355 | 1 | 0.71 | + 6 | 8 | n/a | (S)-8-amino-7-oxononanoate synthase BioU |
| Sll1371 | 1 | 0.7 | − 3.16 | 9 | n/a | SyCrp1 transcription factor |
| **Ssl2245** | **1** | **0.34** | **− 3** | **23** | **n/a** | **Putative antitoxin[57]** |
| **Slr0711** | **0.96** | **0.16** | **− 4.58** | **33** | **n/a** | **7-cyano-7-deazaguanine reductase QueF** |
| Sll1315 | 0.93 | 0.63 | − 12 | 35 | n/a | WD40 repeat protein |
| Sll7087 | 0.93 | 0.39 | − 9.99 | 38 | n/a | CRISPR-Cas protein Cmr4 |
| Slr0670 | 0.93 | 0.33 | − 3.97 | 39 | n/a | Universal stress protein |
| *Ssr3189 | 0.89 | 0.39 | − 9.26 | 44 | n/a | Homolog Asl3888 in *Nostoc* 7120 was validated as RBP in PNK assay[46] |
| Slr0147 | 0.85 | 0.37 | − 3.94 | 53 | n/a | 4-vinyl reductase |
| Slr0782 | 0.85 | 0.28 | − 3 | 55 | n/a | L-amino acid dehydrogenase |
| **Sll0947** | **0.81** | **0.35** | **− 13** | **59** | **n/a** | **Ribosome hibernation factor (Hpf/RaiA/LrtA)** |
| Sll0517 | 0.78 | 0.26 | − 5.3 | 67 | 16 | Cold-induced RRM protein RbpA/Rbp1[49] |
| Ssl3335 | 0.78 | 0.2 | − 12.48 | 72 | n/a | preprotein translocase subunit SecE |
| **Sll0726** | **0.78** | **0.16** | **− 3** | **73** | **n/a** | **Phosphoglucomutase, Pgm** |
| *Slr2130 | 0.74 | 0.3 | − 3 | 76 | n/a | AroB, 3-dehydroquinate synthase |
| Sll1961 | 0.7 | 0.38 | − 12 | 85 | n/a | GntR-type transcriptional regulator[60] |
| Slr1265 | 0.56 | 0.13 | − 4.81 | 129 | n/a | RNAP subunit gamma RpoC1 |
| Slr1030 | 0.44 | 0.28 | − 13 | 174 | 228 | Mg²⁺ chelatase subunit ChlI |

After removing ribosomal proteins, 20 proteins with a promising rank in the RAPDOR analysis are given together with their SVM prediction of RNA binding (Supplementary Data 2). The gene name or locus tag is given, followed by the ANOSIM $R$ value (ANO), the mean JSD, the relative shift in the number of fractions, the absolute rank assigned by RAPDOR and SVM, comments and annotation. n/a indicates an SVM score below the threshold.
Asterisks indicate that the homologs in *Nostoc* 7120 also showed RNA-dependent shifts[46]. See also Figs. 5 and 6 and Supplementary Data 2. Boldface letters indicate RBPs validated in PNK assays (Fig. 7).

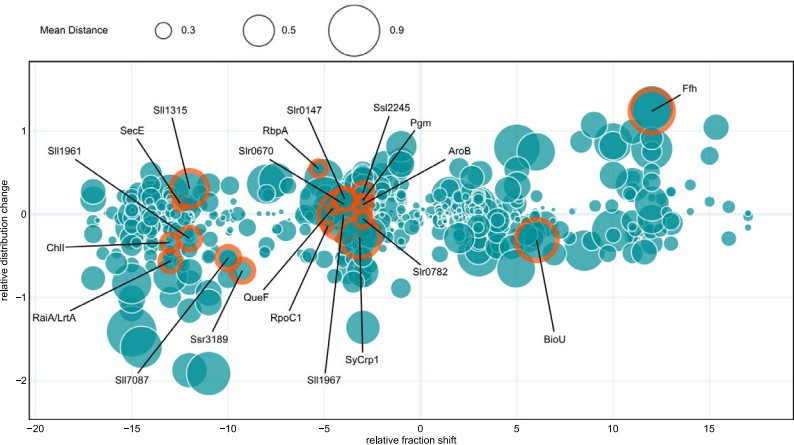

**Fig. 5 | Visualization of shift types from *Synechocystis* Grad-R data.** Each bubble represents a single protein identified by mass spectrometry, the bubble size corresponds to the mean JSD. The x-axis shows the relative fractional shift, while the y-axis shows the entropy gain (positive) or loss (negative) upon RNase treatment. This reflects whether the treatment led to a sharper or broader distribution of the protein (see "Methods"). Proteins that are promising candidates for RNA binding are circled in orange. Ffh is the signal recognition particle protein, Sll1967 is the 23S rRNA (uracil(1939)-C(5))-methyltransferase RlmD, Sll7087 is the type III-Bv CRISPR-Cas protein Cmr4[55], RaiA/Lrt is the ribosome-associated translation inhibitor A[56], RbpA is a cold-induced RRM domain RBP[49], Ssl2245 is a putative antitoxin[57], QueF is a homolog the enzyme 7-cyano-7-deazaguanine reductase involved in biosynthesis of the modified nucleoside queuosine, Ssr3189 is a protein of unknown function conserved in cyanobacteria, its homolog Asl3888 in *Nostoc* 7120 was validated as RBP in PNK assay[46], see Table 1 for further details. Source data are provided as a Source Data file.

weight fractions upon RNase digestion are on the left side of the plot in Fig. 5, while proteins that shifted to higher molecular weight fractions are on the right side. The majority of the selected candidates were located in the upper left quarter, including RbpA. Upon RNase digestion, these proteins shifted to lower molecular weight fractions and

narrowed in distribution. Nine of the selected proteins (RaiA/LrtA, ChlI, Sll1961, Sll7087, Ssr3189, SyCrp1, RpoC1, Slr0782, Sll1967) were located in the lower left quarter, meaning that the distribution along the gradient broadened and the respective proteins shifted to lower molecular weight fractions upon RNase treatment. Two of the selected

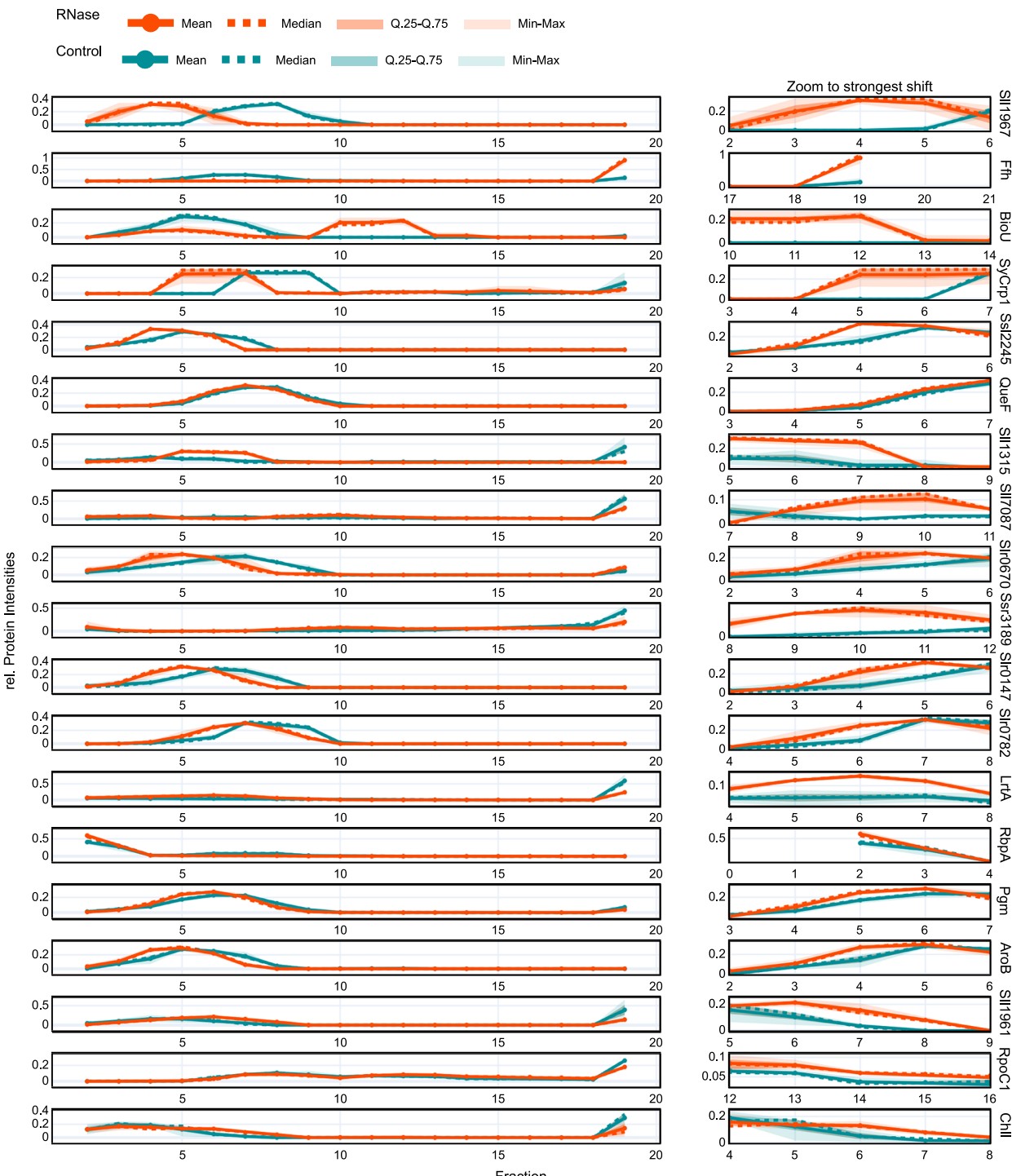

**Fig. 6 | Distribution profiles of proteins shifting in the gradients.** Left panel: global view, right panel: zoom to the strongest shift upon RNase treatment identified via the soft argmax of the position-wise relative entropy to the mixture distribution. Mean (solid line) and median (dotted line) are shown, with the interquartile range shaded and the minimum–maximum range in lighter shading. Source data are provided as a Source Data file.

candidates showed a rightward shift, Ffh located in the upper right quadrant, indicating a sharpened distribution, and BioU (Slr0355) in the lower right quadrant, indicating a broadened distribution. Ffh is a proven RBP[47]. Therefore, its right shift, from fractions 6–8 to the pellet fraction (Fig. 6), likely indicates lowered solubility when the RNA component was lost. BioU is a suicide enzyme involved in biotin synthesis[61]. Its shift from fraction 6 to fraction 12 (Figs. 5, 6) indicates association with a larger molecular complex, but not a drastically lowered solubility as found for Ffh.

The distribution profiles of the selected candidates showed a variety of larger protein shifts, as well as smaller quantities of a respective protein shifting detected by RAPDOR (Fig. 6). For instance, RbpA showed only a small shift, but RAPDOR annotated it with a high ANOSIM *R* value, placing it on rank 67, showing its good accuracy in detecting small shifts. This kind of small shift was also be seen in the distribution of QueF. Most of the protein was present in fractions 6–8, upon RNase digestion only a very small amount of the protein shifted to lower molecular weight fractions. However, the ANOSIM *R* value was

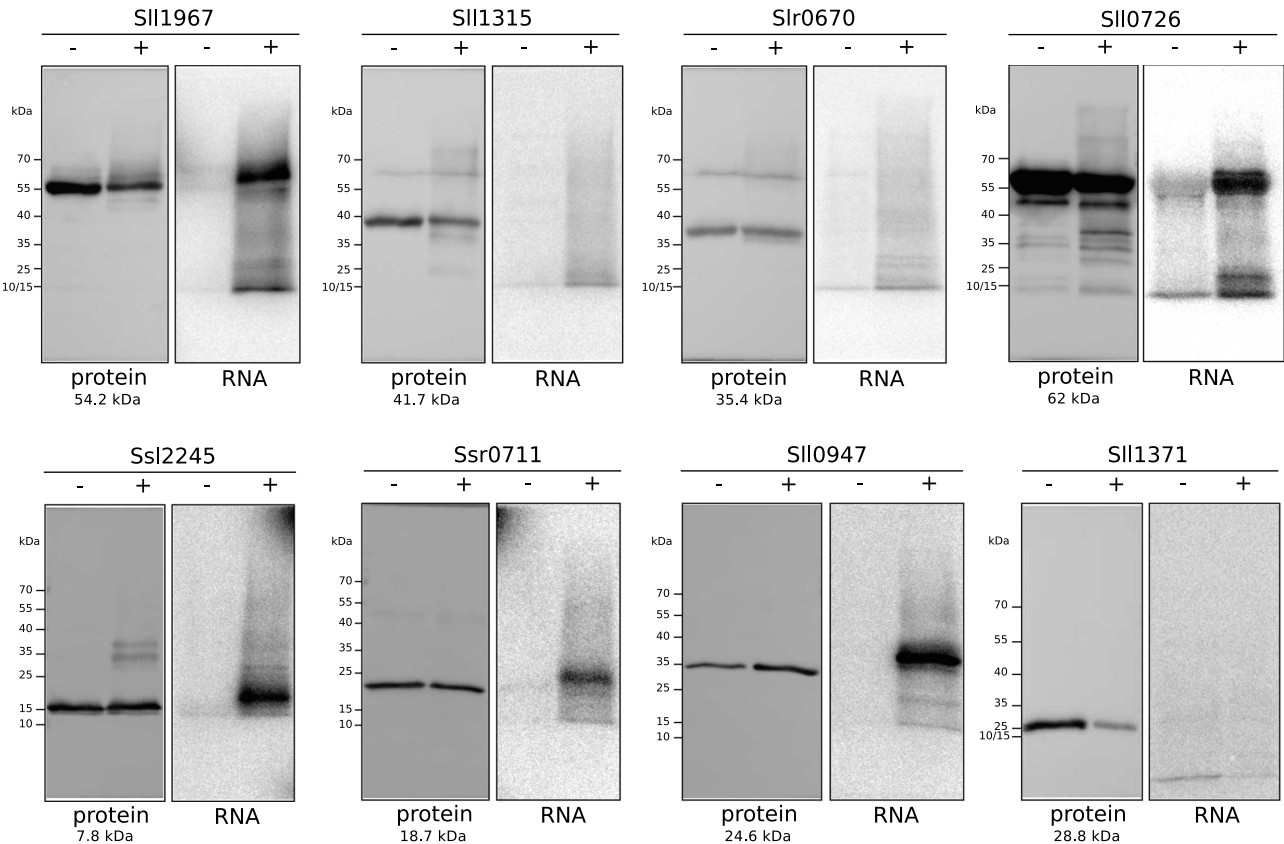

**Fig. 7 | Validation of the GradR screen.** PNK assays of the following candidate proteins are shown: Sll1967 (23S rRNA (uracil(1939)-C(5))-methyltransferase RlmD), Sll1315 (WD40 repeat protein), Slr0670 (universal stress protein), Sll0726 (phosphoglucomutase), Ssl2245 (a putative antitoxin), Slr0711 (preQ(1) synthase homolog QueF), Sll0947 (ribosome-associated translation inhibitor RaiA/LrtA) and Sll1371 (SyCrp1 transcriptional regulator). For each protein, the left panel (protein) shows the respective Western blot membrane, the right panel the autoradiography (RNA) after UV-crosslinking (+), RNase treatment and labeling using PNK and $^{32}$P-ATP or the control without UV-crosslinking (-). The position of size markers is indicated on the left side, the calculated molecular masses of the respective proteins is given underneath. All Western blots were carried out using mouse monoclonal anti-FLAG antiserum conjugated to horseradish peroxidase at a titer of 1:5000. One representative set of results out of two is shown. Source data are provided as a Source Data file.

0.96, meaning that even though the amount of shifting protein was small, it was detectable in most replicate comparisons of the protein.

**Validation of RNA binding in vivo**

We used isotope labeling by PNK[33] as an assay to test the RNA-binding capacity of eight selected proteins in vivo. These proteins were ranked by RAPDOR with ANOSIM *R* values between 1 and 0.78 on positions 2 to 73. One of these eight proteins was predicted with a score ≥ 0.25 as an RBP by TripPepSVM, three were classified as showing a significant shift by R-DeeP (Supplementary Data 2). We generated *Synechocystis* 6803 reporter strains that expressed each candidate protein fused to a 3xFLAG epitope tag under control of the copper-inducible P*petE* promoter from a plasmid vector. Cells were UV-treated to covalently crosslink the RBP candidates to their associated RNA(s), followed by purification of the complexes using magnetic beads binding to the FLAG epitope tag under stringent conditions to disrupt protein-protein interactions. RNA was partially degraded by benzonase to select for nuclease-protected RNA covalently bound to the protein. The remaining RNA fragments then were $^{32}$P-labeled using PNK, and the resulting RNA-protein complexes fractionated by SDS-PAA gel electrophoresis and transferred to a nitrocellulose membrane. The protein-RNA interaction in vivo was confirmed if the protein was detected by anti-FLAG Western blotting and a corresponding radioactive signal was observed in the cross-linked sample. In this assay, five of the eight selected proteins showed clear positive signals (Fig. 7). One of the validated proteins was Sll1967, a putative homolog of the

23S rRNA (uracil(1939)-C(5))-methyltransferase RlmD[50] (Supplementary Fig. 9) that was also shifting significantly in the gradients, classified by RAPDOR on rank 2, supported by a high SVM score and detected also by R-DeeP. The other validated proteins were Ssl2245, a putative antitoxin[57], the preQ synthase QueF homolog, the ribosome-associated translation inhibitor homolog RaiA (Sll0947), and the phosphoglucomutase Pgm (Sll0726). The assay was negative for the SyCrp1 transcriptional regulator Sll1371, the WD40 repeat protein Sll1315 and the universal stress protein Slr0670.

**Application of RAPDOR to spatial proteomics**

To showcase its potential for further applications, we used RAPDOR to re-analyze existing spatial proteomics data. Spatial proteomics is a rapidly evolving field in which the subcellular localization of proteins is addressed. This is especially relevant in the context of many signaling proteins shuttling between the cytoplasm and nucleus, such as Smad complexes crucial for transduction of transforming growth factor β (TGF-β)-superfamily signals from transmembrane receptors into the nucleus[62], or the plant photoreceptor phytochrome for the relay of light signaling[63], the endocytosis of activated receptor proteins in healthy and cancer cells[64], or the movement of moonlighting proteins which perform different functions at different sites[65]. Martinez-Val et. al.[39] investigated proteome dynamics upon Epidermal growth factor (EGF) stimulation in HeLa cells and compared it to unstimulated cells. The entire proteome was assessed across distinct cellular compartments at different time points after stimulation. Each compartment

comprised two fractions: cytosolic, membrane-bound organelles, and the nucleus/nucleolus. Such a setup results in six categories instead of numerical fractions, which is also supported by RAPDOR, but results in slightly different plots. The dataset is suitable for analysis by RAPDOR as it only identifies shifts in probability distributions, independently of whether they stem from a continuous (GradR) or a discrete (spatial proteomics) probability distribution. Only the averaging kernel is disabled when working with discrete distributions, as this assumes that fractions are sequentially connected.

The original analysis pipeline calculated a mobility score per protein[39]. This score reflects the total percentage of shifting protein. In addition, it performs two moderated *t*-tests at the fractions with the highest loss/gain in protein amount, respectively. The resulting two *p*-values are then combined using Fisher's method in order to identify proteins with a significantly different distribution after EGF treatment. Here we show that an analysis using RAPDOR yields similar results when calculating the *JSD* instead of a mobility score, as both measures show a strong correlation (Supplementary Fig. 10). We further used the pre-analyzed data from Martinez-Val et al.[39] as input for RAPDOR. This data was already filtered and imputed. Since the dataset included four replicates per condition, it was possible to calculate *p*-values using RAPDOR's global ANOSIM mode (see Methods). However, since the power of the test is still weak for four replicates, we decided for a less strict cutoff and used an adjusted *p*-value of 0.1, thus accepting a higher false discovery rate. Our results show that the majority of the proteins identified in the original publication also showed a high *JSD* of at least 0.2 and a significant shift when analyzed via RAPDOR at every treatment time point except at 90 min (Fig. 8A). Especially, the shifts of GRB2, CBL, and SHC1 at two and eight minutes after EGF treatment were captured by RAPDOR. Those shifts indicate the recruitment of receptor tyrosine kinase (RTK) adapter proteins, as already highlighted[39]. Interestingly, the analysis pipeline used in the original publication did not detect some clearly shifting proteins, including FOXJ3. This showed a shift at all time points, which was most likely missed because the original publication based the statistical test on selecting the fractions with the highest loss or gain. In contrast, our approach does take all fractions into account by comparing distributions. Here, FOXJ3 accumulated in membrane-bound fractions and reduced its presence in the others (Fig. 8B). This leads to a low loss in the fraction with the largest loss and can result in a false negative observation when using *t*-tests to identify a substantial reduction over all replicates at this fraction. Although nothing is currently known about FOXJ3's involvement in EGF signaling, RAPDOR identified two proteins that have been linked to this pathway. The endocytic scaffolding protein intersectin (ITSN1) and microphthalmia-associated transcription factor (MITF) both showed shifts from the nucleus to cytosolic fractions at 2, 8, and 20 min that were not captured by the original analysis pipeline. While MITF is proposed to negatively regulate epidermal growth factor receptor (EGFR) expression[66], ITSN1 is known to mediate EGFR signaling through modulating its ubiquitylation via the RTK adapter CBL[67].

Another problem might arise from the fact that combining two *p*-values using Fisher's method, as done in the original publication, assumes their independence and is known to produce too low *p*-values for dependent tests. However, since the tests here are based on the same protein distribution, they are highly dependent. On the other hand, using the mobility score cutoff most likely counteracted this problem. In contrast, RAPDOR's global ANOSIM only assumes that the *R* values of different proteins follow the same distribution. This seems to be valid for all four time point comparisons when observing the histograms of *R* values (Supplementary Fig. 5).

These findings indicate the applicability of RAPDOR for experiments different than in-gradient profiles, and the analyzed dataset is a promising starting point for other researchers to detect further protein redistributions. For visualization, all the supplementary JSON files

can be plugged into local versions of RAPDOR and are provided here as Supplementary Data 1.

## Discussion

### Non-parametric peak-independent GradR shift detection

Gradient fractionation of RNase-treated samples to identify RBPs was previously performed with extracts from mammalian cells[27] and subsequently analyzed using the R-DeeP pipeline. This pipeline uses a parametric approach by relying on fitting a multi-Gaussian distribution for each sample and protein. However, its authors are aware of the fact that this might not model all protein distributions equally well, and thus some proteins get tagged as problematic in the final output. In addition, the selection of peaks is essential for this pipeline since the parametric *t*-test is only performed at two positions in the gradient. This approach can lead to issues, as illustrated by the distribution patterns of RpsQ in the *Synechocystis* dataset and TIM50 in the human HeLa dataset (Supplementary Fig. 4 and Fig. 4).

To address these issues, we developed RAPDOR, which can rank such protein profiles well because it does not rely on a parametric approach by fitting peaks or their positions, but rather uses a non-parametric approach of quantifying changes in the distributions. In contrast to R-Deep, it does not use a *t* test, which assumes that the normalized amount of protein follows a Gaussian distribution. While a *t* test is robust against violations of this assumption for large sample sizes, three replicates per group might produce misleading results when assumptions are not met. In contrast, RAPDOR's non-parametric ANOSIM does not rely on assumptions about the underlying data distribution and is therefore more resistant to these problems. However, as is typical for non-parametric tests, this comes at the cost of reduced statistical power. This is, however, compensated by using rankings, as shown below.

### The RAPDOR ranking and selection of R-value cutoffs

A *p*-value cutoff of 0.05 is commonly used because it represents a 5% probability of observing a false positive (Type I error), which is generally considered an acceptable level of risk in many scientific studies. On the other hand, less powerful tests may fail to detect true effects, increasing the risk of false negatives (Type II error), especially when sample sizes are small. As highlighted, the ANOSIM test applied by RAPDOR is likely to suffer from the latter problem for the commonly used sample size of three. This is even the case when using our global *p*-value calculation. However, Clarke et al. already pointed out that the *R* value itself can be used as a comparative measure[43]. Consequently, we implemented a ranking approach based on the *R* value and showed that this ranking tends to place RNA-binding proteins and thus shifting proteins to the top ranks. This is usually sufficient, since evidence gained via high-throughput proteomics is mostly validated using independent follow-up experiments. A user of RAPDOR can therefore pick proteins from the top ranks for such validation.

Defining a universal cutoff for the *R* value, on the other hand, is challenging, as the appropriate threshold may vary across experimental conditions. This is because the distribution of shuffled and original *R* values differs between experiments (Supplementary Fig. 5), largely due to differences in measured noise. Therefore, we decided to determine a cutoff by selecting the 95 percentile of all R-values generated via shuffling treatment labels. Users can adapt this procedure, as we demonstrated its ability to identify previously unknown RBPs in *Synechocystis,* and it captured the majority of known human RBPs. However, it is important to note that this does not allow a user to draw conclusions about Type I or Type II error rates, and we recommend increasing the number of replicates if *p*-value calculation is essential. While we recognize that increasing the number of replicates can be both time-consuming and costly, ongoing advances in proteomics, such as multiplexing techniques like TMT labeling, are steadily

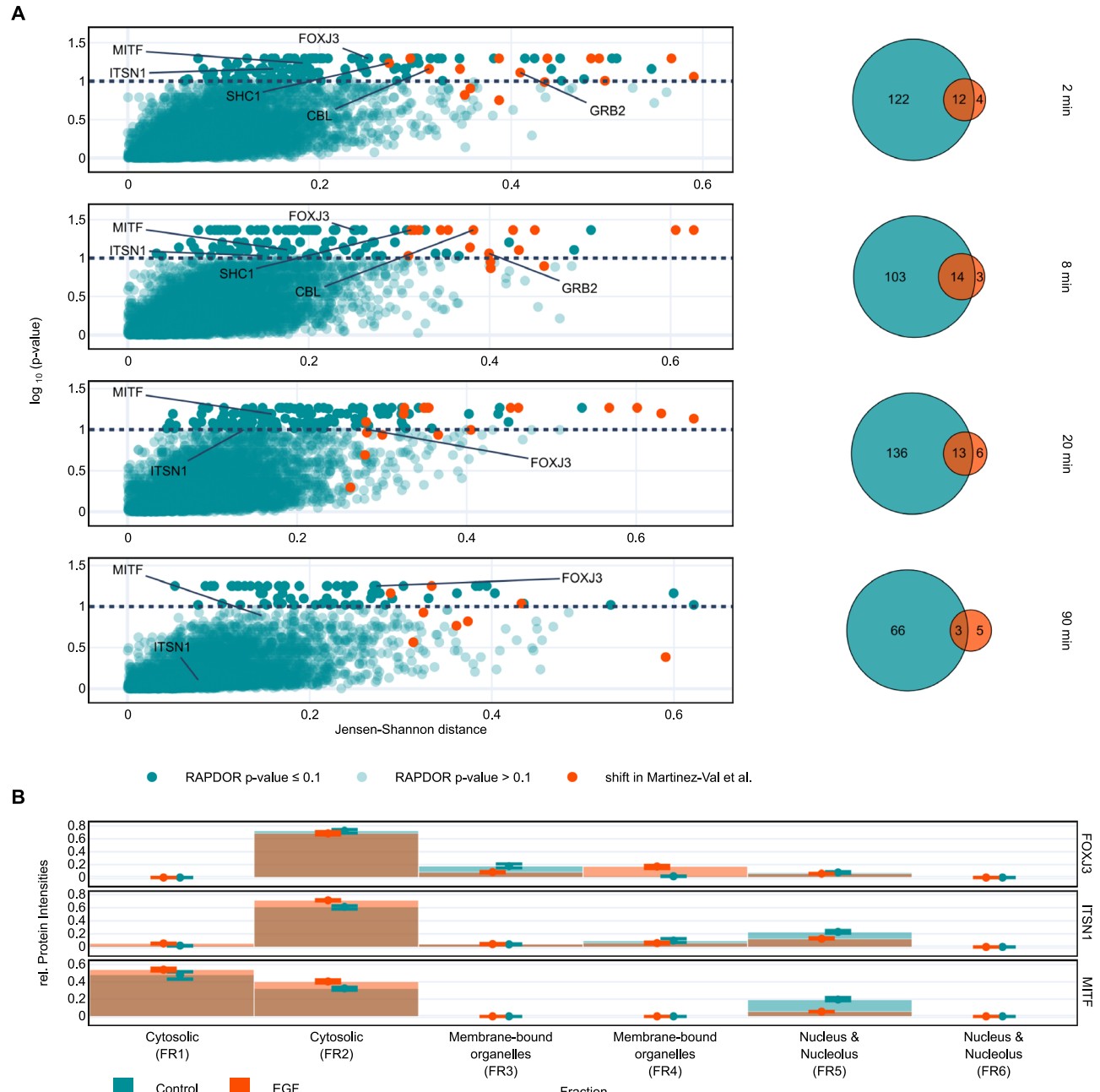

**Fig. 8 | Analysis of spatial protein redistribution upon EGF treatment. A** Left column shows the effect size (Jensen-Shannon distance) vs the -log10 of the adjusted global ANOSIM *p*-value at 2, 8, 20, and 90 min of EGF treatment (*n* = 4). Black dotted lines indicate a typical *p*-value cutoff for such an experiment. Orange dots represent proteins that were identified with a distribution significantly different using the original analysis[39]. The right column shows Venn diagrams comparing proteins with a significant redistribution (*p* < 0.1) called by RAPDOR vs those identified in the original publication. **B** Redistribution of FOXJ3 (*p* = 0.051), ITSN1 (*p* = 0.069) and MITF (*p* = 0.058) at 2 min of EGF treatment (*n* = 4). Dots and bars show the mean, while error bars indicate the interquantile range. After treatment, FOXJ3 showed a significant (Benjamini/Hochberg adjusted ANOSIM *p* < 0.1) accumulation in membrane-bound organelles (FR4). In contrast, ITSN1 and MITF shifted towards cytosolic fractions. These shifts were not captured by the original analysis pipeline[39]. Source data are provided as a Source Data file.

reducing these barriers, making high-replicate experimental designs more feasible.

## Bioinformatic prediction of RBPs in cyanobacteria

The main strategies used for the bioinformatics prediction of RBPs have been based on the identification of RNA-binding domains through comparing structural properties (SPOT-tru[68]), evolutionarily conserved residues and structural properties (SPOT-seq[69]), or sequence similarities and general biochemical properties (RBPpred[17] or Deep-RBPpred[18]). However, recent findings about RBPs have shown that many of them lack canonical RNA-binding domains and instead bind RNA through intrinsically disordered regions. Therefore, approaches that rely on the composition of specific amino acid patterns (such as tripeptides) have emerged as an alternative. The Tri-PepSVM algorithm[19] uses a string kernel support vector machine (SVM) and the composition of specific tripeptides from known RBPs or non-RBPs to make predictions. Unlike other bioinformatics tools, which use eukaryotic and prokaryotic RBPs as a training dataset, the original TriPepSVM workflow enables users to select a specific strain's proteome and the proteomes of evolutionarily related organisms to create

a training set. This is crucial because RBPs in eukaryotes and prokaryotes may not rely on exactly the same features. The ability to use a custom bacterial training dataset, coupled with TriPepSVM's superior performance[19] let us select it for our analyses. In this paper, we used a modified version of TriPepSVM, described in more detail in Brenes-Álvarez et al.[46] that was trained on a custom dataset. In short, we selected proteomes of cyanobacteria with diverse morphologies and lifestyles. In addition to these proteomes, we also included those of two extensively studied Gram-negative bacteria, *E. coli* K12 and *Salmonella typhimurium* LT2. Furthermore, we optimized the specific parameters used by TriPepSVM for this training dataset using a 10-fold cross-validation setting[46].

## RNase sensitive gradient fractionation of *Synechocystis*

Gradient fractionation of RNase-treated samples to identify RBPs was previously performed with extracts from mammalian cells[27], from *Salmonella*[28], as well as from *Pseudomonas aeruginosa* and its bacteriophage ΦKZ[21]. Here we show that the method could successfully be applied to *Synechocystis*, a membrane-rich photosynthetic cyanobacterium.

Popular alternative methods for the identification of RBPs without gradient centrifugation and fractionation, such as the orthogonal organic phase separation[70] and complex capture[71,72] protocols, rely on UV-induced crosslinking of RBPs to RNA in live cells. However, photosynthetic bacteria are rich in pigments that absorb light of various wavelengths, including UV, necessitating high and partially damaging irradiation doses. Moreover, the GradR approach provides information about co-fractionating proteins that could be involved in the same ribonucleoprotein complex. This is exemplified here for ribosomal proteins and our finding of Sll0947 as the cyanobacterial homolog of the ribosome hibernation factor (Hpf/RaiA/LrtA), which was co-fractionating with ribosomal proteins in fraction 19. We verified the RNase-induced shifts that are crucial for this method by analyzing ribosomal proteins of the small and large subunit (Fig. 3). The analysis of the entire dataset benefitted from the here developed RAPDOR workflow that allowed peak-less shift identification.

We are aware of the fact that the RAPDOR workflow may also miss potential RNA-dependent proteins as false negatives, especially when using strict cutoff values for the ANOSIM *R*. This is illustrated here by Rps1A and Rps1B encoded by *slr1356* and *slr1984*, respectively. Rps1 is thought to participate in recruiting mRNA to the 30S subunit[73], but homologs exist only in some bacteria. Rps1A and Rps1B in *Synechocystis* 6803 are involved in the Shine–Dalgarno-independent initiation of translation[74,75] and therefore are known RNA-dependent proteins. Rps1b was captured by RAPDOR, on rank 105, while Rps1a was placed on rank 186, with an ANOSIM *R* value of 0.44, below our cutoff.

Despite some exciting examples of RBPs detected by RAPDOR, the total number of RBPs remains likely gravely underestimated. For example, the majority of proteins annotated with the RNA-binding GO term were not experimentally verified here. The reasons certainly include a lack of resolution. We estimate that the molecular mass difference required for a protein to shift by one fraction is at least 50 kDa (Supplementary Fig. 3). Thus, only proteins binding to RNA at least 150 nt in length can shift by one fraction if this RNA is entirely degraded. Another important aspect may be that only a fraction of an RBP was actually bound to RNA, while a larger fraction was not. Finally, some proteins sharing the GO annotation simply had an insufficient number of replicates with a signal in the mass spec data. In contrast, the fact that TriPepSVM detected all the annotated RBPs likely indicated bias from training. GO-annotations in *Synechocystis* mainly stem from homology inference, which is also the kind of data the SVM model was trained on.

## Identifying RBPs in cyanobacteria

Since it has become increasingly clear in recent years that especially enzymes involved in large metabolic pathways may possess moonlighting abilities to function as RBPs in addition to their enzymatic function[76–78], we included eight enzymes with a promising ANOSIM *R* value in the list of candidates. These proteins are putative RNA modifying enzymes, such as the RNA methyltransferase RlmD and the queuosine synthesis protein QueF; an enzyme involved in amino acid metabolism (the L-amino acid dehydrogenase Slr0782), and one involved in glycolysis (Pgm), which were placed by RAPDOR on ranks 2, 33, 55 and 73. The other four enzymes are BioU, involved in biotin synthesis, a vinyl domain-containing enzyme of unknown functionality, AroB, an enzyme involved in amino acid biosynthesis, and the Mg$^{2+}$ chelatase subunit ChlI on ranks 8, 53, 76 and 174 (Table 1).

One of the most promising candidates with a distinctive shift (Fig. 6) was Sll1967, a homolog of RlmD or RlmCD, which are involved in the methylation of 23S rRNA in *E. coli*, *Streptococcus pneumoniae* and *Bacillus subtilis*[79–81], as is indicated by the conservation of all 5 cysteine residues which bind a 4Fe-4S center, form the catalytic site and residues binding the SAM methyl group donor (Supplementary Fig. 9A, B). However, experimental validation in cyanobacteria was lacking. We now demonstrate its pronounced RNA-binding activity (Fig. 7), which points at a role as an RNA chaperone in addition to its RNA methyltransferase activity. Even more intriguing was the validation of Slr0711/QueF as an RBP because this 7-cyano-7-deazaguanine reductase catalyzes the conversion of preQ0 to preQ1, i.e., it performs a step in the cofactor biosynthesis, but is not supposed to interact with RNA as a substrate[82]. Some RNA modifiying enzymes, such as the tRNA methyltransferase TrmA and the tRNA pseudouridine synthase TruB have previously been demonstrated to act also as RNA chaperones[83,84]. Thus, Sll1967 and Slr0711 discovered here as strong RNA binders are candidates for further research. Moreover, queuosine biosynthesis has recently been characterized as relevant for the regulation of lifestyle decisions in Gram-positive and Gram-negative bacteria[58].

Another interesting RBP identified here is Sll0947, initially described in *Synechococcus* sp. PCC 7002 as Lrt, expressed from a rapidly induced gene when the cells were transferred from light to darkness[85]. Later, this observation was extended to include *Synechocystis* 6803[56], where the association of Sll0947 with ribosomes was demonstrated, as was its relevance in survival after stress exposure[86]. Structural modeling predicts this protein to be a homolog of the ribosome-associated translation inhibitor RaiA (Supplementary Fig. 9C, D). Work in enterobacteria suggested RaiA (39/59% identical and similar residues with Sll0947; Supplementary Fig. 9C) as inactivating 70S ribosomes and storing them as so-called sleeping ribosomes[87], consistent with its observed expression profile in cyanobacteria[56,85]. Here, we showed that Sll0947 is indeed RNA binding (Fig. 7).

Of great interest are also the two remaining RBPs identified in this work. *Synechocystis* 6803 encodes with *sll0726* and *slr1334* two potential phosphoglucomutases. The *sll0726* product (Pgm) is highly homologous to other bacterial phosphoglucomutases and was suggested to represent a target of thioredoxin regulation[88], while the enzyme encoded by *slr1334* was predicted to encode a phosphogluco/phosphomannomutase bifunctional enzyme[89]. Recently, Pgm has gained attention as a key regulatory point in the metabolism of carbon storage compounds, especially regarding the carbon flux between glycogen and the central carbon metabolism[90], as it was shown to provide ~99% of the phosphoglucomutase activity[91]. Our data here show that Pgm/Sll0726 is an RBP and therefore likely a moonlighting enzyme.

Ssl2245, here characterized as an RBP, is a putative antitoxin. While it is associated with the PemK-type toxin Sll1130, it could not be assigned to any known antitoxin family[57]. However, PemK toxins frequently function as endoribonucleases[92,93], therefore, future work may

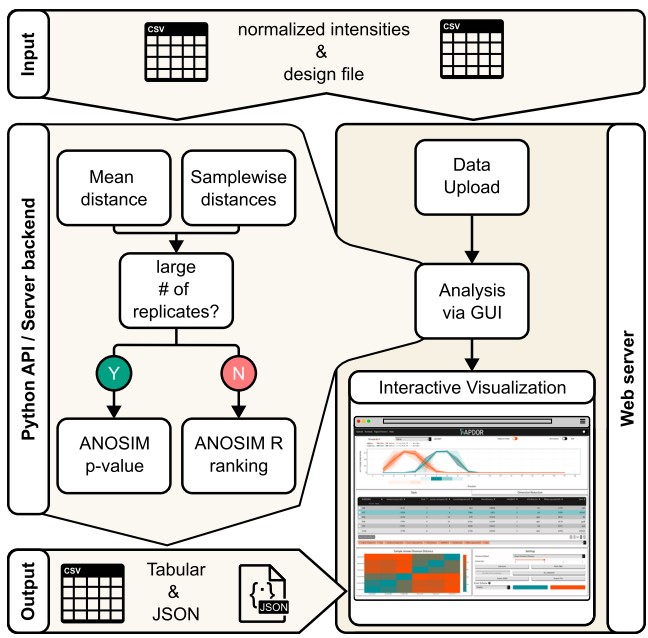

**Fig. 9 | RAPDOR supports multiple usage methods.** RAPDOR can conduct analyses either directly in Python or as an interactive web server. The Python-analyzed data can be exported as a JSON file and displayed in the web server environment afterwards.

address the Sslr2245 function as binding RNA, e.g., a possible substrate for the toxin Sll1130.

## The RAPDOR workflow is broadly applicable

One focus of our research was to develop a tool that is easily applicable for similar experiments. As a result, the tool is broadly applicable and can detect shifts in proteome distribution between two conditions in experiments involving multiple fractions. As the tool compares the same protein under different conditions, it expects raw mass spectrometry intensity sums as input. However, common proteomics practices, such as data imputation, can enhance the performance of RAPDOR. Importantly, RAPDOR employs a non-parametric statistical approach that does not rely on assumptions about the underlying distribution of the profiles. To the best of our knowledge, the non-parametric ANOSIM has not been applied in this context before. In both GradR data and in the spatial proteomics example, the previous analysis was either based on a parametric modeling of the distribution itself or involved parametric tests like the Student's *t* test on derived values. In addition, implementational aspects are important for a broad applicability. Hence, the RAPDOR pipeline is optimized for low runtime and memory consumption, making it suitable for average consumer laptops. Further, the tool is supported by comprehensive online documentation for the Python API and the Dash graphical user interface (GUI), featuring tutorials on setting up the tool with custom datasets. Such datasets can be analyzed either locally via Python or in the GUI directly (Fig. 9). It is then possible to set up a server to host such pre-analyzed data via a configuration and a JSON file. This disables the computationally more expensive analysis and provides a direct way to visualize data along with a publication. It also allows for more in-depth and visual analysis of individual distribution changes between different conditions. Consequently, we set up a web server to display the *Synechocystis* GradR data under the following link: https://synecho-rapdor.biologie.uni-freiburg.de.

To demonstrate the potential to distribute pre-analyzed data, we analyzed publicly available datasets, which focus on the redistribution of proteins across cellular compartments of human HeLa cells. We offer these datasets in JSON format, compatible with the RAPDOR

visualization tool, facilitating their utilization by other researchers for hypothesis generation (Supplementary Data 1).

## Methods

### Culture conditions

Triplicate cultures of *Synechocystis* 6803 PCC-M[94] were maintained in 100 mL BG11[95] supplemented with 20 mM 2-{[1,3-Dihydroxy-2-(hydroxymethyl)propan-2-yl]amino}ethane-1-sulfonic acid (TES), pH 7.5, under continuous white light of 50 μmol photons $m^{-2}$ $s^{-1}$ at 30 °C.

### Cell lysis and RNA removal

Cells were harvested after reaching an $OD_{750}$ of 0.9 by centrifugation (4000 × *g*, 4 °C, 20 min). The cell pellet was resuspended in 800 μL lysis buffer + 1 mM DTT (20 mM Tris/HCl (pH 7.5), 150 mM KCl, 10 mM $MgCl_2$) containing protease inhibitor (complete EDTA-free protease inhibitor, Roche). Cell lysis was performed by mechanical disruption using a pre-chilled Precellys homogenizer (Bertin Technologies). To remove unlysed cells and glass beads, samples were centrifuged (500 × *g*, 2 min, 4 °C), and the supernatants collected for further processing. To solubilize membrane proteins, cell lysates were incubated in the presence of 2% n-dodecyl β-D-maltoside for 1 h in the dark at 4 °C. Cell debris was then removed by centrifugation (21,000 × *g*, 4 °C, 1 h). The cleared lysate was divided into two fractions of equal volume. One of the fractions was incubated with 100 μL RNase A/T1 mix (Thermo Fisher Scientific) for 20 min at 22 °C. The control fraction was treated with 100 μL mock buffer (50 mM Tris-HCl (pH 7.4), 50% (v/v) glycerol) + 0.4 U of RNase inhibitor (Ribo-Lock, Thermo Fisher Scientific) for 20 min at 22 °C. Samples were then kept on ice until loading.

### Gradient preparation and fractionation

Gradients were prepared using the Gradientmaster 108 (Biocomp) to obtain a linear gradient of solution 1 (10% (w/v) sucrose in lysis buffer) and solution 2 (40% (w/v) sucrose in lysis buffer). Open-Top Polyclear™ Centrifuge Tubes 9/16 × 3-1/2 in. (Seton Scientific) were used as centrifugation tubes. Gradients were overlaid with 400 μL of solution 1 before loading 500 μL lysate. Separation of the lysate was achieved by ultracentrifugation using a swinging-bucket rotor (Beckman SW40 Ti) for 16 h at 285,000 × *g*. 20 fractions of equal volume (~600 μL) were collected using the PGF ip Piston Gradient Fractionator (Biocomp), except for the pellet fraction (fraction 20), which was collected manually.

### Sample preparation and details of mass spectrometry measurements

One hundred μL of each fraction was used for mass spectrometric analysis of gradient samples (3 biological controls each; 120 samples in total). Samples were mixed with 4 x sample volume of 50 mM triethylammonium bicarbonate (TEAB) and reduced by addition of 0.5 μM Tris(2-carboxyethyl)phosphine (TCEP) followed by incubation at 37 °C for 45 min under constant shaking at 900 rpm. The samples were then alkylated with 0.5 μmol iodoacetamide (15 min, room temperature, dark). Proteolytic digest was achieved by adding 1 μg trypsin (Promega) and incubating the samples for 16 h at 37 °C, 900 rpm. The digestion was stopped by adding 1/26 volume of 50% (v/v) trifluoroacetic acid. Peptides were purified after digestion using Pierce C18 Tips (Thermo Fisher Scientific) according to the manufacturer's protocol. For fractions with the highest protein content (1–10 and 20), the purifying procedure was repeated twice, and the eluates were pooled. Recovered peptides were dried and resuspended in 20 μL of 0.1% acetic acid using an ultrasonic bath. To monitor reproducibility of LC-MS runs, retention time calibration peptides (iRT, Biognosys) were spiked in a 1:100 ratio.

LC-MS/MS analyses were performed on an LTQ Orbitrap Velos Pro (ThermoFisher Scientific, Waltham, MA, USA) using an EASY-nLC II

liquid chromatography system. Tryptic peptides were subjected to liquid chromatography (LC) separation by loading them on a self-packed analytical column (OD 360 µm, ID 100 µm, length 20 cm) filled with 3 µm diameter C18 particles (Dr. Maisch, Ammerbuch-Entringen, Germany). Peptides were eluted by a binary nonlinear gradient of 2–99% acetonitrile in 0.1% acetic acid over 88 min with a flow rate of 300 nL/min and subsequently subjected to mass spectrometry (MS). For MS analysis, a full scan in the Orbitrap with a resolution of 30,000 was followed by collision-induced dissociation (CID) of the twenty most abundant precursor ions. MS2 experiments were acquired in the linear ion trap.

Database searches were performed against all proteins predicted to be encoded in the *Synechocystis* 6803 chromosome (NC_000911.1) and the four plasmids pSYSA, pSYSG, pSYSM, and pSYSX (NC_005230.1, NC_005231.1, NC_005229.1, and NC_005232.1, respectively). The database was supplemented with sequences of known sORFs and RNase A/T, resulting in a total of 3743 entries. Database search as well as label-free protein quantification was performed using MaxQuant (version 2.0.3.0)[96]. Common laboratory contaminants and reversed sequences were included by MaxQuant. Search parameters were set as follows: trypsin/P specific digestion with up to two missed cleavages, methionine oxidation and N-terminal acetylation as variable modifications, carbamidomethylation at cysteines as fixed modifications, match between runs with default parameters enabled. The mass tolerance for matching of precursor ion was 4.5 ppm and was set to 0.5 Da for fragment ions. The minimum peptide length was specified with 7 amino acids. The FDRs (false discovery rates) of protein and PSM (peptide spectrum match) levels were set to 0.01. Two identified unique peptides were required for protein identification. LFQ[97] and iBAQ[98] were exported as quantitative values of protein abundance. The DAPAR R package was used to impute missing values in fractions where a protein showed a signal in two out of three replicates. To not produce a signal for very different profiles, we only used data imputation for the proteins with the 95 percent lowest mean JSD within the same treatment. The generated MS data have been deposited to the ProteomeXchange Consortium[99] via the PRIDE partner repository[100] with the dataset identifier PXD045848.

## Polyacrylamide gel electrophoresis and Western blotting
20 µL of each fraction was boiled with 1x protein loading buffer (Tris/HCl 0.5 M (pH 6.8) 50 mM, SDS 2%, glycerol (v/v) 6%, DTT 2 mM, bromophenol blue (w/v) 0.01%) at 95 °C for 10 min and then separated by 15% SDS-polyacrylamide gel electrophoresis (SDS-PAGE)[101]. For membrane transfer of smaller proteins (< 25 kDa), 1 mA/cm² was used. Larger proteins (> 28 kDa) were separated on 12% SDS-PAGE and blotted at 1.5 mA/cm². Mouse monoclonal anti-FLAG antiserum conjugated to horseradish peroxidase (ANTI-FLAG® M2-Peroxidase,# A8592 Sigma-Aldrich) at a titer of 1:5000 was used in Western blots.

## RNA isolation and Northern blotting
RNA extraction and northern blotting were performed as described previously[25]. Primer and oligonucleotide sequences are listed in Supplementary Table 1. Signals were visualized using Typhoon FLA 9500 (GE Healthcare) and Quantity One software (Bio-Rad).

## Construction of recombinant *Synechocystis* 6803 strains
To express tagged RBP candidate proteins in *Synechocystis* 6803, the universal plasmid for subcloning X-54 was constructed. To generate the backbone, a segment of pUC19 was amplified using primers P1 and P2 (Supplementary Table 1). As insert, the P*petE* promoter, *sll7087*, 3 x FLAG tag and *oop* terminator were PCR-amplified from plasmid V-37 using primers P3 and P4. Both fragments were treated with *Dpn*I (Thermo Fisher Scientific), assembled using AQUA cloning[102] and transformed into *E. coli* TOP10F' cells (Thermo Fisher Scientific).

Plasmid X-54 then was used as a PCR template with primers P5 and P6 to reverse amplify P*petE*, pUC19, 3xFLAG and *oop*, leaving out the gene *sll7087*. All other genes of interest (goi) were then inserted into this backbone. The genes were amplified from *Synechocystis* 6803 gDNA using primers P7/P8 (*sll1967*), P9/P10 (*sll1371*), P11/P12 (*ssl2245*), P13/P14 (*slr0711*), P15/P16 (*sll1315*), P17/P18 (*slr0670*), P19/P20 (*sll0947*) or P21/P22 (*sll0726*). The fragments were assembled using AQUA cloning and transformed into TOP10F'. The P*petE*-GOI-3xFLAG-*oop* fragment of all isolated plasmids was amplified with primers P23/P24. Plasmid pVZ322s[103] was digested with *Xmn*I (NEB) and treated with FAST AP (Thermo Fisher Scientific). The linear plasmid and the insert fragments P*petE*-GOI-3xFLAG-*oop* were assembled using AQUA cloning[102]. They were subcloned into TOP10F. All PCR reactions were performed with PCRBIO HiFi Polymerase (PCR Biosystems). Sanger Sequencing Economy Run (Microsynth) was used to verify the correct sequence of all plasmids. *Synechocystis* 680*3* was transformed with isolated plasmids by electroporation according to Kraus et al.[104] with the following modifications: ice-cold *ultrapure* water was used to wash the cells, and only 200 ng of each plasmid were used. Gentamicin (5 µg/mL) was used for selection.

## Test of RNA-binding capacity
To verify direct RNA-protein interactions in vivo, the PNK assay[105] was performed as developed by Brenes-Alvarez et al.[46] with the following modifications. P*petE*-controlled protein expression was induced in 250 mL *Synechocystis* 6803 cultures at an OD$_{750}$ of 0.7–0.8 by the addition of 0.31 µM Cu$_2$SO$_4$. After 24 h, cultures were transferred to a 21 × 14.5 × 5.5 cm³ plastic tray on ice and cross-linked three times with UV of 254 nm at 0.75 J/cm$^{-2}$ in a UV Stratalinker 2400 (Stratagene) with gentle shaking. Negative controls were not cross-linked and were also stored on ice. The samples were prepared and separated by SDS-PAGE. After western blotting, radioactive ink (highly diluted [γ-$^{32}$P]-ATP) was used to label the membrane contours.

## Jensen-Shannon Distance
The Jensen-Shannon Distance (*JSD*, see Eq. 3) is the default effect size measure for an individual replicate of the RAPDOR tool. For two probability distributions *P* and *Q*, the *JSD* is defined as follows:

$$M(x) = \frac{1}{2} \times (P(x) + Q(x)) \tag{1}$$

$$D(P||Q) = \sum_{x \in X} P(x) \log\left(\frac{P(x)}{Q(x)}\right) \tag{2}$$

$$JSD(P||Q) = \sqrt{\frac{D(P||M) + D(Q||M)}{2}} \tag{3}$$

Here, *M* refers to the mixture distribution of *P*,*Q* and *D* is known as the Kullback-Leibler (KL) divergence (see Eqs. 1 and 2)), which is a commonly used measurement on probability distribution. In contrast to the more prominent KL divergence, which is not symmetric, the *JSD* is a metric function. This makes the *JSD* suitable for the downstream analysis for a non-parametric test of similarities between conditions (see the section on the ANOSIM *R* value below).

## Expected position of the strongest shift
For each treatment *t* the position of the strongest shift $S_t$ is determined using the position-wise relative entropy, which has its origin in convex programming (Eq. 4). We used it since its sum is equal to the KL divergence, given that both *P* and *M* are probability functions[106]. In more detail, it is based on likelihood ratios of either treatment and the mixture distribution at position *x*, weighted by the probability P(x) of observing fraction *x*. To determine the position of the strongest shift,

we use a localization operator (Eq. 5) that corresponds to the temperature-scaled soft-argmax function, which is often used in machine learning. It counteracts uncertainty in our data and performed better than the argmax function for distributions with broader peaks. Basically, the soft-argmax function returns the expected position of the strongest shift. $\beta$ is the hyperparameter for temperature scaling and must be carefully chosen for a given dataset.

The relative fraction shift (Eq. 6) is then obtained via a subtraction of the expected position of the strongest shift of the control samples ($t = -$) from the RNase-treated one ($t = +$).

$$w_t(x) = \begin{cases} P(x)\log_2(P(x)/M(x)) & P(x) > 0, M(x) > 0 \\ 0 & M(x) \geq 0, P(x) = 0 \\ \infty & otherwise \end{cases} \quad (4)$$

$$S_t = \sum_{x \in X} \frac{e^{\beta w_t(x)}}{\sum_{x \in X} e^{\beta w_t(x)}} \cdot x \quad (5)$$

$$RPS = S_+ - S_- \quad (6)$$

### Non-parametric test analysis of similarities (ANOSIM)

The non-parametric test analysis of similarities (ANOSIM) calculates a test statistic called ANOSIM $R$ via a ranked distance matrix[43]. This is achieved via labeling fields in this matrix with two distinct labels, depending on whether the distance originated from two samples of the same treatment $W$ or a different treatment $B$. The $R$ value for $n$ samples is then calculated via the following equation:

$$R = \frac{\overline{r_B} - \overline{r_W}}{(n(n-1)/2)/2} \quad (7)$$

Hereby $\overline{r_B}$ is the average rank of the different treated fields (e.g., in GradR control vs RNase), and $\overline{r_W}$ refers to the average rank of fields from the same treatment (e.g., in Grad RNase vs. RNase or control vs control). Via shuffling of the labels ($B$, $W$), it is possible to estimate a distribution over the $R$-values and thus calculate a $p$-value.

While ANOSIM is a well-known statistical test for community ecologists, the procedure can in principle be adapted to any kind of multivariate data analysis, as long as there is a sufficient number of replicates per condition. However, three replicates are not sufficient to achieve a $p$-value below a significance level $\alpha \leq 0.05$. This is because the number of permutations given a balanced sample layout with two conditions follows Eq. 8, thus resulting in only 10 distinct permutations for $n = 3$.

$$\frac{(2n)!}{2(n!)^2} \quad (8)$$

Depending on the number of replicates and instances (here, proteins), it is possible to calculate $p$-values using the $R$ value distribution of all instances as background. This method, further termed the global mode, assumes that $R$ values of different instances follow the same distribution. While this may not always be the case, such assumptions are generally robust and are employed by other tools. For example, the differential expression tool limma utilizes information from all genes for its underlying moderated $t$ test[107]. Further, the ANOSIM test statistic is bound between $[-1, 1]$. This makes it comparable between proteins with the same number of samples, while it is not influenced by the effect size due to its non-parametric nature. Values close to 1 show that the distances between the same treatment are lower than the distances between different treatments. In contrast,

$-1$ shows the opposite, and values around zero indicate evenly distributed distances. This offers the possibility to rank proteins according to a decreasing $R$ value to identify those with reproducible shifts.

### Reanalysis of R-DeeP dataset

Raw mass spectrometry files from the R-DeeP experiment in HeLa cells were downloaded from ProteomeExchange (PXD010119). Proteins were identified using MaxQuants database search running on default parameters with the human proteome downloaded from Uniprot (UP000005640). Resulting Intensities were subsequently used as an input for the original R-DeeP script. To focus solely on comparing the statistical methods and minimize any effects caused by R-DeeP's normalization, the R-DeeP normalized intensities were used as input for RAPDOR.

### Runtime & memory consumption benchmarking

The runtime and memory consumption analysis was performed on a single Intel i5-10210U CPU core. In addition, a dataset with more replicates was simulated by duplicating the existing measurements from the *Synechocystis* 6803 GradR experiment, resulting in a dataset with nine replicates each. This was used exclusively for runtime and memory consumption benchmarking.

### Support vector machine classification of candidate RBPs

In addition to the experiment-based approaches, we used TriPepSVM[19], to predict candidate RBPs. TriPepSVM relies on the idea that a support vector machine (SVM) can make a prediction on whether a protein is an RBP solely based on amino acid sequence. The basic principle is creating positive and negative datasets, which will then train the SVM, leading it to predict RBPs based on sequence alone. A protein sequence is cut up into overlapping k-mers, which form a vector. This vector is analyzed by the learning algorithm, and tripeptides are scrutinized by occurrence within the protein. This leads the SVM to make a prediction based on the likelihood of said protein belonging to the RBP or non-RBP category. We have trained TriPepSVM for cyanobacterial genomes and applied it to two use cases, on *Nostoc* 7120[46] and on *Synechocystis* 6803 (this work). To train the algorithm for cyanobacterial RBP candidates, the predicted set of proteins encoded in 10 cyanobacteria, *E. coli* K12, and *Salmonella typhimurium* LT2 was downloaded from the UNIPROT database[108]. To avoid duplicate annotations or paralogs, CD-Hit[109] was utilized to create a unique dataset and remove redundant proteins with a similarity > 90%[109]. This was done for the 12 considered organisms, yielding 1151 unique RBP candidates that were merged into a positive dataset.

From the pool of remaining proteins, all proteins with an RNA-binding domain in the Pfam database or with annotation keywords related to nucleic acid binding in UNIPROT, or GO terms in QuickGO, were discarded. After filtering by CD-Hit, a negative dataset of 33,860 unique non-RBP was obtained. The kmerPrediction.r script of TriPepSVM[19] was modified for the implementation of the SVM by changing the package used in conjunction with "KeBABs"[110] from "e1071" to "LiblineaR". For the selection of the best combination of parameters, each dataset was randomly split into training (90%) and testing (10%) samples and used in a 10-fold cross-validation by randomly sampling the subsets. The parameter combination resulting in the largest average balanced accuracy (BACC) was selected. We used a positive class weight of 2.7, a negative class weight of 0.05, cost = 1 and k-mer = 3. For further details, see ref. 46. The proteins encoded by all 3.681 annotated protein-coding genes in *Synechocystis* 6803 were scored, and a conservative threshold of 0.25 was selected as the SVM score for the classification as potential RBP. This prediction yielded a list of 306 candidate RBPs in *Synechocystis* 6803 (Supplementary Data 2).

## Reporting summary

Further information on research design is available in the Nature Portfolio Reporting Summary linked to this article.

## Data availability

The mass spectrometry datasets generated in this study have been deposited at the ProteomeXchange[99] Consortium (http://proteomecentral.proteomexchange.org) via the PRIDE partner repository[100] under the identifier PXD045848 (Analysis of RNA-dependent proteins in the cyanobacterium *Synechocystis*). Previously generated mass spectrometry data are available under the identifier PXD010119 (The concept of "RNA dependence" - Proteome-wide and quantitative identification of protein interactions dependent on RNA[27]) and PXD023690 (Spatial-proteomics reveals phospho-signaling dynamics at subcellular resolution[39]). The processed data for *Synechocystis* 6803 accessibility and visualization are available at: https://synecho-rapdor.biologie.uni-freiburg.de. Source data are provided in this paper.

## Code availability

• The RAPDOR tool is available as a pypi package and its documentation[111] is hosted on https://domonik.github.io/RAPDOR/ • The code used to analyze the data, including the modified R-DeeP script[112] is available as a Snakemake workflow on GitHub: (https://github.com/domonik/synRDPMSpec).

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

## Acknowledgements

We appreciate the support by the Deutsche Forschungsgemeinschaft (DFG, German Research Foundation) through the research training group "BioInMe" 322977937/GRK2344 to L.H., D.R., R.B., and W.R.H., DFG priority program SPP2002 "Small Proteins in Prokaryotes, an Unexplored World" (grants HE 2544/12-2 to WRH, BA 2168/21-2 to R.B., and BE 3869/5-2 to DöB) and grant HE 2544/22-1 to W.R.H. MBA was granted by an Alexander von Humboldt postdoctoral fellowship. This study was supported by the German Research Foundation (DFG) under Germany's Excellence Strategy (CIBSS - EXC-2189 - Project ID 390939984). It was also funded by the Deutsche Forschungsgemeinschaft (DFG, German Research Foundation) – Project ID 499552394 – SFB 1597. We thank the Bio 3 IT department for their help in setting up the web server.

## Author contributions

W.R.H. and L.H. designed the research. The experiments were performed by L.H. and V.R. S.M., P.G., J.B., and D.B. carried out the mass spectrometry and related analyses. H.R. and M.B.A. applied the

TripepSVM approach, and D.R. and R.B. developed the RAPDOR tool. All authors analyzed the data. W.R.H., L.H., R.B., and D.R. wrote the manuscript with input from all authors. All authors read and approved the final manuscript.

## Funding

## Competing interests
The authors declare no competing interests.
