## [Transparent Peer Review file · Nature Communications]

RAPDOR: Using Jensen-Shannon Distance for the computational analysis of complex proteomics datasets

Corresponding Author: Dr Wolfgang Hess

Version 0:

Reviewer comments:

Reviewer #1

(Remarks to the Author)

In this paper, Hemm et al present RAPDOR, a non-parametric analysis tool for studying distribution-based proteomics data. They combine this tool with GradR to identify RNase-driven shifts in proteins' gradient profiles as a way of discovering novel RBPs. They employ this in a cyanobacterium, *Synechocystis* sp. PCC 6803, and propose several new RBP candidates, although none of these were validated experimentally. They found that the complement of proteins with significant shifts exhibited relatively little overlap between their analysis method and the R-DeeP analysis pipeline and noted that RAPDOR was able to identify shifts not picked up by the other analysis tool. They also highlight the greater flexibility and 'user friendliness' of RAPDOR. To this end, they provide two more, surface-level examples of where RAPDOR could be applied in the context of intracellular fractionation experiments. Despite a focus on the value of this analysis tool, there was not sufficient validation or benchmarking to deem it an improvement on other available software. Nor was there sufficient exploration of the GradR dataset for it to provide new insight into the RBPome of cyanobacteria. The paper would need to refocus either on an in-depth exploration of the RAPDOR analysis method, or on a more detailed study of the GradR dataset for it to offer a meaningful contribution to the field.

Major comments

- A major focus of the paper is the density gradient fractionated proteomic dataset generated in *Synechocystis* sp. PCC 6803. However, the paper is lacking in a global overview of this dataset. There are no quality control plots of the proteomic data, making it impossible to have confidence in the validity of claims made. A global overview of the dataset would also improve the paper's biological insight. Recent studies relying on gradient fractionation include holistic analyses of whole dataset behaviour, alongside plotting individual protein profiles. Geladaki et al (2019, doi: 10.1038/s41467-018-08191-w), for example, used dimensionality reduction approaches to study subcellular protein distribution. Such an approach could be a valuable orthogonal way of understanding how proteins group and how they behave following RNase treatment. The authors might consider incorporating PCA into the RAPDOR tool, for example, as an additional facet to the analysis pipeline.
- There is very little exploration of how the putative RBPs identified in this study compare to what is already known about RBPs generally and, more specifically, RBPs in bacteria. There have been diverse efforts to study RBPs in bacteria, including some studies in cyanobacteria. Among these, it looks like there is a bioRxiv paper from some of the same authors studying a different cyanobacterium (doi: <https://doi.org/10.1101/2024.04.03.587981>). Where direct comparison is possible (i.e., in relatively closely related species), RBPs identified in the present dataset should be compared to those identified in other studies (such a comparison is mentioned in the text but results should be displayed). Where species are not closely related, it would still be of value to determine whether there might be homologous or orthologous RBPs. Additionally, determining whether the identified proteins have the general properties of RBPs, such as in amino acid composition and isoelectric point, would be helpful. I am assuming that the use of TriPepSVM is meant to be a proxy for this but it isn't sufficiently well explained in the paper to make it informative. Additionally, given how small the overlap of predicted RBPs is with those that were experimentally identified, other comparisons would strengthen the paper's conclusions.
- Further to the above point, there is also no validation of whether identified candidates can bind RNA, or even whether they would be likely to bind to RNA, based on their physicochemical properties. This is particularly important, given that this

paper is claiming that RAPDOR can identify very subtle shifts in distribution profiles. To determine whether identifying such subtle shifts is important, the authors would need to employ a method such as the PNK assay to assess whether candidates have RNA-binding activity. This would be a necessary validation of the RAPDOR analysis approach.

- The comparison of RAPDOR to other analysis methods was not explored in sufficient depth to be informative. The comparison of the RAPDOR method with the R-Deep analysis pipeline did not systematically assess the relative benefits of the two approaches. While RAPDOR appears to work better for some proteins, are there others for which R-Deep is better? The size of this dataset makes it difficult to objectively assess the relative merits of the methods. The authors should assess relative performance in the dataset from the original Caudron-Herger paper, where many more shifts were observed to explore differential discovery potential. For the spatial proteomics datasets, the comparison between RAPDOR and the original analyses are very unclear. While comparisons are referenced in the text, they are either not interpretable from the figures included, or entirely missing from the figures. If clearer and more systematic comparisons were included, it would make it much easier to understand what the benefits of RAPDOR are and would make a more compelling case for users to choose this approach over others.

Minor comments

- Lines 94-96

The text and schematic in Figure 1A suggest that the shifts in proteins' elution profiles following RNase digestion are a result of loss of the mass of RNA. Our understanding of the method is that the shift can also be caused by the disruption of RNA-bridged protein complexes, meaning that proteins that exhibit shifts can be either directly binding to RNA or can be RNA-dependent through their presence in RNA-containing complexes. This should be clarified in the schematic and text.

- Line 184

Details of the fractionation were given in a supplementary table. Why not plot this information alongside a representative gel to give an indication of consistency between replicates?

- Lines 167-174

There is discussion of RNA pattern in the untreated sample but no reference to the pattern in RNase treated samples. Would it be possible to include either an RNA gel following RNase treatment or a bioanalyser run to assess the extent of RNA digestion? This would be particularly useful in differentiating between a technical or biological explanation for how few proteins were observed to change upon RNase treatment.

- Lines 175-178

There should be a global overview of the proteomics data in the figures, beyond a description in the text. E.g., Is there good clustering and/or correlation between replicates? What is the overall distribution of intensities across the fractions and does it reflect that observed in protein gels?

- Figure 2

This is a nice schematic of the workflow but doesn't necessarily justify being a full main figure. Perhaps this would be better as a supplementary figure or could be included in a larger figure in a simplified form. This is also true for the paper's final figure.

- Lines 322-324

Is the fact that there are relatively few shifts among the large ribosomal subunit surprising? Has this been observed in other datasets or is it specific to this one? At first glance, this would seem to suggest that there is something wrong with the data, such as incomplete RNA digestion, so this would need to be better addressed in the text.

- Lines 388-389

How the choice of cut-off was made is not adequately explained or justified. Perhaps a figure showing where known RBPs sit within the ranking would be helpful. Does this R threshold encompass all RBPs? Given how few RBPs have been classified in cyanobacteria, perhaps it would make more sense to set this threshold based on a human dataset, where there is extensive data on classification of RBPs. If this tool is to be used by the community, there would need to be evidence that this threshold is valid under other conditions.

As discussed in the general comments above, there is very little insight into whether the threshold set here actually yields proteins with the properties of RBPs. Given the arbitrary nature of the threshold selected, this would be an important addition to this figure.

- Lines 344-346

TriPepSVM's inclusion here, and the information that it is providing, needs to be better described. The very small overlap between predicted and true RBPs does not feel as if it is sufficiently addressed. The reason given in the text is that there might be novel RNA binding properties but we would still expect that many of the RBPs in cyanobacteria do have canonical RBP properties. How well does this prediction algorithm perform generally? Can we trust that it is accurately predicting the full host of RBPs?

- Line 354

Again, as mentioned above, would it be possible to include comparison to other available datasets from cyanobacteria or bacteria generally?

- Lines 413-463

This section feels as though it would be more appropriate for the discussion than the results, since there is no validation of their RNA-binding activity in this paper.

- Lines 466-476

It is not clear why this specific dataset was selected, or why the RAPDOR analysis method is appropriate for a type of data which is fundamentally different from data that consists of sequential fractions. This would need to be explained better.

- Lines 483-484

The text states that the JSD and mobility score calculations yielded similar results. There is no evidence in the figures to show this – a plot would need to be added with this comparison.

- Lines 489-491

The text states that the results are recapitulated but the figure is not intuitive and the result is not well explained. A more explicit assessment of this, such as a venn diagram or bar chart displaying numbers of consistent/ discordant results might be informative.

• Lines 510-512

This statement does not feel justified as there is not sufficient validation that the additional hits identified by RAPDOR are biologically meaningful. There would need to be a much more comprehensive comparison of the results from the two methods, as well as experimental validation of the importance of additional hits for this claim to be appropriate.

• Lines 521-531

It is not clear why this further dataset is included in this paper, as it is addressed in such a cursory manner. There is no overview of what the analysis with RAPDOR looked like for this data, nor is there a clear description of what the data is. The text also states that the main findings of the publication are reproduced by RAPDOR but there is only one panel in a supplementary figure of the results from this analysis and this doesn't offer any form of comparison. This only indicates replication of one of the paper's findings. If others were recapitulated, these would also need to be shown.

(Remarks on code availability)

Reviewer #2

(Remarks to the Author)

(Remarks on code availability)

Reviewer #3

(Remarks to the Author)

A non-parametric tool for comparing fractionated mass-spec data is a very useful and exciting addition to the RNA/protein field. It may also be useful for analysing gradient-fractionated polysomal profile of RNAs and proteins between two conditions. Since the focus of this study is the development of an analysis tool, I do not have enough bioinformatic background to assess the tool at the coding level. However some questions regarding the wet-lab aspects and the data may benefit from more discussions:

1. Lysis without RNase inhibitor will cause some RNA-protein complex to dissociate during lysis. The ladder signals of individual transcripts in the Northern blot in Fig.S1 suggest they are actively being degraded, without addition of exogenous RNases.
2. Lysis without cycloheximide will cause ribosome run-off, leaving proteins involved in active translation out from the dataset. eg. One may expect more large subunit ribosomal proteins to appear as RNA-binding if cycloheximide was present.
3. If the lysate consists of the total ribonucleoprotein in the cell, why were only 31% of the annotated proteome detected (Line 178)? Is this a loading or detection issue or is the cell stage used in this study known for expressing only 31% of the proteome?
4. Presence of individual proteins/complexes in the sucrose gradient should follow a distribution spanning multiple fractions, but they appear to be single dots in Fig.S2. What do the dots represent? Mean or median fraction number? This is not clearly described in either main text or in legend.
5. Not all ribosomes are actively translating (such as the case described in Line 425), so some ribosomal proteins may be at heavy fractions and only some of the same protein (from ribosomes in contact with mRNA) move to lighter fractions following RNase treatment. Is the authors' algorithm sensitive enough to pick up such case? Can this explain the lower number of ribosomal proteins classed by RAPDOR as RNA-binding compared to TripPepSVM (Line 348)?
6. RNA-binding proteins can also be detected by comparing proteins co-purified with RNA with/without UVC crosslinking (OOPS or TRAPP approach). TRAPP dataset for human HEK293T cells is available (<https://doi.org/10.1093/nar/gkad1249>). Has a RNase-gradient-based dataset been done on HEK cells? Or has UV-crosslinking approach been done on cyanobacterium? If so how do the RNA-binding proteins compare between the approaches?
7. Typo "her" in Line 535.
8. In theory the RAPDOR tool may also be able to analyse mass-spec data across a time course, instead of fractions, between two conditions. Has this been tested? The TRAPP time course dataset above may be a suitable test data.

(Remarks on code availability)

Reviewer #4

(Remarks to the Author)

In their study, Wolfgang R. Hess and Rolf Backofen introduce RAPDOR, a novel computational analysis method tailored for proteomic datasets, particularly those stemming from gradient fractionation-based proteomics aimed at identifying RNA-binding proteins (RBPs). The core of RAPDOR lies in its utilization of the Jensen-Shannon Distance (JSD) and Analysis of Similarities (ANOSIM), augmented by additional computations such as relative entropy and Shannon entropy. As a proof-of-concept, the authors analyze data from *Synechocystis* 6803, uncovering novel RBP candidates when compared to the original GradR analysis. Furthermore, they apply RAPDOR to spatial proteomics, revisiting proteomics data from various cellular compartments, and conclude by discussing potential applications of their method, re-analyzing proteomics data from different cellular compartments. At last they discussed the potential applications of RAPDOR.

This study slightly changes the original analysis pipeline of GradR and R-Deep, mainly by introducing nonparametric analysis instead of parametric analysis. This might be of great importance in statistical theory, but its importance and novelty in biological contexts remain somewhat questionable. Besides, the outcome difference between different analysis methods is not sufficiently validated. In summary, this article would be better to be published in a journal focusing more on statistics or computation. If the article is otherwise considered for publication, the following questions and concerns must be thoroughly addressed.

Major:

1. What is the outcome difference between RAPDOR and the original analysis pipelines in GradR and R-Deep? A thorough and comprehensive analysis between different methods need to be presented in your article.
2. What is your final criteria in determining whether a protein is an RBP? The selection of your cutoff should at least be mentioned and validated with regard of other established datasets.
3. There are many RBP datasets generated with different methods, especially for *Mus musculus* and *Homo Sapiens*. A comparative analysis integrating your study's results, the original results, and these established datasets is necessary.
4. To bolster the biological significance of the findings, biochemical experiments should be conducted to:
(1) Confirm that one or more of the newly identified RBPs are indeed RBPs.
(2) Verify that one or more RBPs detected by the original analysis pipeline but not by RAPDOR are false positives.

Minor:

1. Please clarify the types of data that can be analyzed using RAPDOR by providing additional examples or explanations to broaden its application scope more distinctly.

(Remarks on code availability)

Version 1:

Reviewer comments:

Reviewer #1

(Remarks to the Author)

Firstly, we would like to thank the authors for their thoughtful responses to our comments and commend them for the effort they have put in to addressing our concerns. The paper benefits from the improved exploration of the nature of proteins identified and from better comparison to existing datasets. However, even with the added experiments and analysis, we still feel that the strength of this paper is in introducing a new, more flexible analysis tool, rather than offering substantial biological insight. Therefore, we feel that the paper might be better suited to a more technical or bioinformatics focused journal.

A few specific comments that we don't feel were fully understood or addressed are also included below:

- Our suggestion of incorporating dimensionality reduction methods to understand the data holistically was meant to refer to plotting individual protein behaviour patterns, rather than just samples. This is not essential to the paper's message but might be an interesting extension in future.
- The addition of the PNK assay is very helpful for knowing whether this pipeline can identify new RBPs. However, it is not clear how R-Deep versus RAPDOR performs in identifying new RBPs (were the three from R-Deep those that had RNA-binding activity?). Also, as we suggested previously, this would be much more compelling if 'candidates' with very subtle shifts could be validated to indicate that these shifts are meaningful.
- The use of TriPepSVM over other RBP prediction tools is still not adequately justified. Perhaps it would be improved by comparing results across different tools or providing more insight into the general features of identified proteins that make them likely to be RBPs.

(Remarks on code availability)

Reviewer #2

(Remarks to the Author)

(Remarks on code availability)

Reviewer #3

(Remarks to the Author)

This revision by Hemm et al addressed most of reviewers comments, and the manuscript is much better supported by experimental validations and clearer descriptions of its supported data type and limitations.

However one of my original questions was whether there is a UV-crosslinking-based dataset for RNA-interactome (eg. Oops or TRAPP), of which the stringency of RBP identification can be compared with this RNase-gradient-based method. In authors' rebuttal they cited Brenes-Álvarez et al., 2025, which did not have a UV-crosslinking-based RNA interactome, but a CLIP-Seq dataset which looked at protein-interacting RNAs, not RNA-interacting proteins, and therefore cannot be compared with the authors' dataset. A quick search online did not yield any articles applying UV-crosslinking-based methods in cyanobacterium.

That said, research articles using human cells can be found in abundance, including the HeLa cell UV-crosslinking-based dataset in <https://doi.org/10.1016/j.cell.2018.11.004> and the HeLa cell RNase-gradient-based dataset in <https://doi.org/10.1016/j.molcel.2019.04.018>, which was extensively cited in authors' article. It is a shame that this opportunity to compare between the two approaches is missed. The method described by the authors requires 40 fractions to be analysed in mass spectrometer per sample (w/wo RNase) per replicate. This is very costly compared to the UV-crosslinking-based method, with only two MS injections required (w/wo crosslink) using single-shot label free DIA. Some UV-based methods (like Oops) also selects RNA-binding proteins based on their susceptibility to RNase. Therefore unless this RNase-gradient-based approach provides far superior sensitivity or accuracy for RBP detection, its appeal to other researchers (20x MS cost!) will be limited.

Since I did not see other reviewers writing about the app, I have done a test run in both the latest Linux/python environment and in Windows executable. In both environments the app installed and ran with no issues. The API documentation in <https://domonik.github.io/RAPDOR/> appeared to be still under construction, so I cannot comment on this part. My only issue is the index webpage misses a clear link to its main GitHub page at <https://github.com/domonik/RAPDOR/>. This link is only visible at the installation page. Without this link in the index page will potentially make it more difficult for users to report bugs or incompatibility issues, especially in the future when python or other runtime environments are updated.

Based on its comparison with R-Deep pipeline, I believe the RAPDOR pipeline is solid and the field will benefit from the publication of this non-parametric analysis approach for fractionated data. However without proper justification for the cost of data generation, hence the requirement for this pipeline analysing fractions, its impact in the RNA-interactome field may not be sound.

(Remarks on code availability)

App test-ran under both Linux/python environment and in Windows executable. Installation and execution were fine. The provided test files were successfully analysed. The installation guide and tutorial were user friendly, however the API documentation is still under construction at this stage.

Reviewer #4

(Remarks to the Author)

I appreciate the authors' efforts in providing additional supportive experiments and more comprehensive discussions. In this revised version, the authors have satisfactorily addressed most concerns raised. I believe this manuscript advances the methodology for RNA-dependent protein identification and validates it with wet lab experiments. Therefore, I recommend its acceptance by Nature Communications.

(Remarks on code availability)

Point-to-point replies to the reviews of our manuscript Ref: Nature Communications manuscript,

Authors: Hemm, Rabsch, et al. "RAPDOR: Using Jensen-Shannon Distance for the computational analysis of complex proteomics datasets".

REVIEWER COMMENTS

Reviewer #1 (Remarks to the Author):

In this paper, Hemm et al present RAPDOR, a non-parametric analysis tool for studying distribution-based proteomics data. They combine this tool with GradR to identify RNase-driven shifts in proteins' gradient profiles as a way of discovering novel RBPs. They employ this in a cyanobacterium, *Synechocystis* sp. PCC 6803, and propose several new RBP candidates, although none of these were validated experimentally. They found that the complement of proteins with significant shifts exhibited relatively little overlap between their analysis method and the R-DeeP analysis pipeline and noted that RAPDOR was able to identify shifts not picked up by the other analysis tool. They also highlight the greater flexibility and 'user friendliness' of RAPDOR. To this end, they provide two more, surface-level examples of where RAPDOR could be applied in the context of intracellular fractionation experiments. Despite a focus on the value of this analysis tool, there was not sufficient validation or benchmarking to deem it an improvement on other available software. Nor was there sufficient exploration of the GradR dataset for it to provide new insight into the RBPome of cyanobacteria. The paper would need to refocus either on an in-depth exploration of the RAPDOR analysis method, or on a more detailed study of the GradR dataset for it to offer a meaningful contribution to the field.

Many thanks for your positive and encouraging comments! Following your advice, we have introduced benchmarking to compare it with the only other software available, R-DeeP, using the dataset published with their pipeline. In addition, we have explored the RBPome of cyanobacteria more deeply as outlined below.

Major comments

- A major focus of the paper is the density gradient fractionated proteomic dataset generated in *Synechocystis* sp. PCC 6803. However, the paper is lacking in a global overview of this dataset. There are no quality control plots of the proteomic data, making it impossible to have confidence in the validity of claims made. A global overview of the dataset would also improve the paper's biological insight. Recent studies relying on gradient fractionation include holistic analyses of whole dataset behaviour, alongside plotting individual protein profiles. Geladaki et al (2019, doi: 10.1038/s41467-018-08191-w), for example, used dimensionality reduction approaches to study subcellular protein distribution. Such an approach could be a valuable orthogonal way of understanding how proteins group and how they behave following RNase treatment. The authors might consider incorporating PCA into the RAPDOR tool, for example, as an additional facet to the analysis pipeline.

We appreciate your comments on adding quality control plots and pointing at the publication by Geladaki et al. (2019).

We have now added a new Figure providing the quality control data for the *Synechocystis* 6803 GradR dataset (new Supplementary Fig. S2). This figure presents PCA of IBAQ values as a dimensionality reduction approach (panel B), but we think that the Spearman correlation of IBAQ values between each sample combination (panel A) and the histograms of Spearman correlation of sample-wise individual protein distribution profiles in panel C are more informative.

Following your suggestion, we integrated PCA into the RAPDOR library and added an option to visualize it in the online tool. However, we found that PCA primarily distinguishes proteins based on their peak positions rather than their shifting behavior. As a result, we decided to retain the bubble plot as the default visualization, as it provides a more holistic and intuitive overview of our dataset.

We have added a new Supplementary Fig. S2 and added in the Results (lines 190–196 in the version with tracked changes): “*Nevertheless, all samples showed good correlation (Spearman correlation coefficients of Intensity-Based Absolute Quantification (IBAQ) values ≥ 0.89 between replicates from the same treatments, and ≥ 0.85 between replicates from different treatments), and the principal component analysis revealed a clear separation of RNase-treated and control samples already in the first principle component (24.32 % variance explained; Supplementary Fig. S2).*”

- There is very little exploration of how the putative RBPs identified in this study compare to what is already known about RBPs generally and, more specifically, RBPs in bacteria. There have been diverse efforts to study RBPs in bacteria, including some studies in cyanobacteria. Among these, it looks like there is a biorXiv paper from some of the same authors studying a different cyanobacterium (doi: <https://doi.org/10.1101/2024.04.03.587981>).

We do compare to information on RBPs globally and also specifically in bacteria, especially cyanobacteria now.

We directly compare to the findings from the mentioned publication (Brenes-Álvarez et al., 2025) for another cyanobacterium, *Nostoc* sp. PCC 7120, on multiple occasions. We do so in the Results section on “*Candidates for proteins interacting with RNA from GradR data*”. We also refer to the respective homologs in *Nostoc* sp. PCC 7120 that also showed RNA-dependent shifts in Table 1.

However, it should be kept in mind that *Synechocystis* and *Nostoc* are very different organisms. *Synechocystis* is unicellular and possesses 3,681 annotated protein-coding genes, while *Nostoc* is multicellular, develops multiple different cell types and possesses 5,995 annotated protein-coding genes. In addition, while we here applied laboratory standard growth

conditions, low nitrogen conditions were used in the work by Brenes-Álvarez et al. (2025).

Moreover, we have added comparative information now to Table S3, column S, also referring to further studies.

Where direct comparison is possible (i.e., in relatively closely related species), RBPs identified in the present dataset should be compared to those identified in other studies (such a comparison is mentioned in the text but results should be displayed). Where species are not closely related, it would still be of value to determine whether there might be homologous or orthologous RBPs. Additionally, determining whether the identified proteins have the general properties of RBPs, such as in amino acid composition and isoelectric point, would be helpful. I am assuming that the use of TriPepSVM is meant to be a proxy for this but it isn't sufficiently well explained in the paper to make it informative. Additionally, given how small the overlap of predicted RBPs is with those that were experimentally identified, other comparisons would strengthen the paper's conclusions.

Thanks for this great suggestion. We have added the direct comparison between homologous proteins in the new Supplementary Figure S8. This new Figure contains information on homologs to the RNA modification enzyme SII1967 and the likely ribosome inhibitor SII0947.

- Further to the above point, there is also no validation of whether identified candidates can bind RNA, or even whether they would be likely to bind to RNA, based on their physicochemical properties. This is particularly important, given that this paper is claiming that RAPDOR can identify very subtle shifts in distribution profiles. To determine whether identifying such subtle shifts is important, the authors would need to employ a method such as the PNK assay to assess whether candidates have RNA-binding activity. This would be a necessary validation of the RAPDOR analysis approach.

Many thanks for this suggestion! Following your advice, we have chosen eight predicted RNA-dependent proteins and performed PNK assays. Only one of these eight proteins was predicted by TripPepSVM with a score ≥ 0.25 as an RBP, three were classified as showing a significant shift by R-DeepP. The results of these PNK assays clearly support direct RNA binding for five of these eight proteins. For details, please see our new section "*Validation of RNA binding in vivo*".

We have added a new section on "*Validation of RNA binding in vivo*" to the Results including the new Fig. 8, added a corresponding section in the Materials and Methods, and adapted the text in the Discussion and other chapters to refer to these new results accordingly.

- The comparison of RAPDOR to other analysis methods was not explored in sufficient depth to be informative. The comparison of the RAPDOR method with the R-Deep analysis pipeline did not systematically assess the relative benefits of the two approaches. While RAPDOR appears to work better for some proteins, are there others for which R-Deep is better? The size of this dataset makes it difficult to objectively assess the relative merits of the methods. The authors should assess relative performance in the dataset from the original Caudron-Herger paper, where

many more shifts were observed to explore differential discovery potential. For the spatial proteomics datasets, the comparison between RAPDOR and the original analyses are very unclear. While comparisons are referenced in the text, they are either not interpretable from the figures included, or entirely missing from the figures. If clearer and more systematic comparisons were included, it would make it much easier to understand what the benefits of RAPDOR are and would make a more compelling case for users to choose this approach over others.

Following this advice, we have compared RAPDOR more deeply to the R-Deep pipeline. We benchmarked their performance for RBP and RDP classification and pinpointed strengths and weaknesses of either of the tools for individual protein distributions.

Please see the new paragraph in the results section “*Enhanced RBP classification performance with RAPDOR*” and the new Supplementary Fig. S6A in which we compare the AUROC and AUPRC for RBP and RDP classification of both tools. In the new Fig. 5 and Supplementary Fig. S6B we now provide a Venn diagram on the number of proteins identified by the two approaches to allow an improved direct comparison. And in Fig. S6C we provide the numerical values for the performance measures of RBP and RDP classification for the RAPDOR and R-DeeP tool.

We have improved the paragraph about spatial proteomics by adding the new Supplementary Fig. S9 in which we compare the mobility scores calculated on basis of Jensen-Shannon distances or according to Martinez-Val. et. al. (2021).

Minor comments

- Lines 94-96

The text and schematic in Figure 1A suggest that the shifts in proteins' elution profiles following RNase digestion are a result of loss of the mass of RNA. Our understanding of the method is that the shift can also be caused by the disruption of RNA-bridged protein complexes, meaning that proteins that exhibit shifts can be either directly binding to RNA or can be RNA-dependent through their presence in RNA-containing complexes. This should be clarified in the schematic and text.

Yes, absolutely. Clarification done as suggested.

Fig. 1A was complemented by a protein released from a directly RNA-binding protein upon RNase digestion and we clarified this in the text accordingly (lines 169–171) and in the legend to Fig. 1A.

- Line 184

Details of the fractionation were given in a supplementary table. Why not plot this information alongside a representative gel to give an indication of consistency between replicates?

Yes, this is an excellent idea. We had included such representative protein gels already in the previous version in Fig. 1C and D. To give an indication

of consistency between replicates, we have revised this figure, and entered the information previously in Table S2 now as panel F to Fig. 1.

Fig. 1 was extended and the text in the Results section modified accordingly.

- Lines 167-174

There is discussion of RNA pattern in the untreated sample but no reference to the pattern in RNase treated samples. Would it be possible to include either an RNA gel following RNase treatment or a bioanalyser run to assess the extent of RNA digestion? This would be particularly useful in differentiating between a technical or biological explanation for how few proteins were observed to change upon RNase treatment.

Done as suggested. We added an RNA gel following RNase treatment as panel C now to Fig. 1. One can clearly see that all RNA was digested in fractions 1 to 10 and that all high molecular weight RNA was degraded in fractions 11 to 20. In the last fractions, some RNA molecules remained, mainly <400 nt in length. Some remaining RNA was especially apparent in fraction 20 indicating some protective effects of the co-fractionating ribosomal proteins.

We newly added panel C now to Fig. 1 and refer to it in the Results (lines 180–184).

- Lines 175-178

There should be a global overview of the proteomics data in the figures, beyond a description in the text. E.g., Is there good clustering and/or correlation between replicates? What is the overall distribution of intensities across the fractions and does it reflect that observed in protein gels?

Done as suggested. We added different quality control plots to the supplement (new Supplementary Figure S2). Each of them indicates good quality of our data. Additionally, we provide a global comparison of treated versus untreated samples directly in RAPDOR.

- Figure 2

This is a nice schematic of the workflow but doesn't necessarily justify being a full main figure. Perhaps this would be better as a supplementary figure or could be included in a larger figure in a simplified form. This is also true for the paper's final figure.

We feel the presentation of the workflow in Fig. 2 is very important for the reader. It provides an intuitive overview over the entire approach including its main novelties, setting the stage for all following information. Therefore, we would like to keep Fig. 2 in the main part of the manuscript.

Our final figure is the only figure in which we present the application of RAPDOR to another experimental dataset, the re-analysis of spatial protein redistribution. Therefore, we would prefer to keep also this figure in the dataset. No change.

- Lines 322-324

Is the fact that there are relatively few shifts among the large ribosomal subunit surprising? Has this been observed in other datasets or is it specific to this one? At first glance, this would seem to suggest that there is something wrong with the data, such as incomplete RNA digestion, so this would need to be better addressed in the text.

Yes, it is a typical effect, many proteins of both the large and the small ribosomal subunit do not shift as impressively as RplA and we have observed this behavior also in another study (Brenes-Álvarez et al., 2025). We assume that partial ribosomal subunit complexes remain because they are also stabilized by protein-protein interactions or because some rRNA fragments were unreachable for the RNase treatment, consistent with remaining RNA fragments in fraction 20 as shown now in Fig. 1C.

We have reworked our discussion of these effects in the manuscript in lines 341–345.

- Lines 388-389

How the choice of cut-off was made is not adequately explained or justified. Perhaps a figure showing where known RBPs sit within the ranking would be helpful. Does this R threshold encompass all RBPs? Given how few RBPs have been classified in cyanobacteria, perhaps it would make more sense to set this threshold based on a human dataset, where there is extensive data on classification of RBPs. If this tool is to be used by the community, there would need to be evidence that this threshold is valid under other conditions.

As discussed in the general comments above, there is very little insight into whether the threshold set here actually yields proteins with the properties of RBPs. Given the arbitrary nature of the threshold selected, this would be an important addition to this figure.

We explain the selection of the cutoff now in more detail and show that this procedure captured the majority of human RBPs in the R-DeeP dataset (lines 379–445).

Additionally, we added a section in the Discussion on how to use the ranking and the R value, see “*The RAPDOR ranking and selection of R-value cutoffs*”.

- Lines 344-346

TriPepSVM’s inclusion here, and the information that it is providing, needs to be better described. The very small overlap between predicted and true RBPs does not feel as if it is sufficiently addressed. The reason given in the text is that there might be novel RNA binding properties but we would still expect that many of the RBPs in cyanobacteria do have canonical RBP properties. How well does this prediction algorithm perform generally? Can we trust that it is accurately predicting the full host of RBPs?

TripPepSVM-generated a list of 306 candidate RBPs, among them were alone 47 of the 52 ribosomal proteins. Because these were all detected by MS, we could discuss that, based on the experimental evidence, 29 ribosomal proteins were classified by RAPDOR as RBP (and 4 by R-DEEP). Hence, RAPDOR provided an overlap of more than 60% with the TripPepSVM prediction, which is a pretty good result given the resolution limits of the gradient fractions.

A major reason for the otherwise relatively small overlap between predicted and confirmed RBPs lies in the fact that only about 31% of the proteome was detected by MS. We refer to this point in line 188.

However, for the most interesting class of proteins, those that were classified by RAPDOR as RBP and not predicted by TripPepSVM, indeed the answer is that they have properties not previously identified as RNA-binding. Here it is very relevant that only a single of the seven experimentally validated RBPs was also predicted by TripPepSVM, strongly supporting our claim. Please see our new section “*Validation of RNA binding in vivo*” in the Results and the accordingly revised section “*Novel RBPs in cyanobacteria*” in the Discussion.

- Line 354

Again, as mentioned above, would it be possible to include comparison to other available datasets from cyanobacteria or bacteria generally?

Yes, another dataset is available for the cyanobacterium *Nostoc* 7120 (Brenes-Álvarez et al., 2025). We now directly compare to the findings from this publication on multiple occasions. We do so in the Results section on “*Candidates for proteins interacting with RNA from GradR data*”. We also refer to the respective homologs in *Nostoc* 7120 that also showed RNA-dependent shifts in Table 1 and we have added such information also in Table S3, column S, also referring to further studies.

- Lines 413-463

This section feels as though it would be more appropriate for the discussion than the results, since there is no validation of their RNA-binding activity in this paper.

We fully agree with this statement. Section matching to previous lines 413–463 has been thoroughly revised and transferred to the Discussion as a new sub-chapter “*Novel RBPs in cyanobacteria*”.

- Lines 466-476

It is not clear why this specific dataset was selected, or why the RAPDOR analysis method is appropriate for a type of data which is fundamentally different from data that consists of sequential fractions. This would need to be explained better.

We revised the first paragraph of this section to better explain why this specific dataset was selected. We chose this dataset to showcase that RAPDOR can be applied to different types of data and chose especially spatial proteomics as a rapidly evolving field causing a wide interest.

In detail, we now explain that only the averaging kernel of the RAPDOR tool assumes sequential fractions. By disabling the kernel for categorical fractions, RAPDOR is applicable to spatial proteomics as it can be represented as a discrete probability distribution.

- Lines 483-484

The text states that the JSD and mobility score calculations yielded similar results. There is no evidence in the figures to show this – a plot would need to be added with this comparison.

An additional plot (Fig. S9) was added to the supplement. This plot shows the strong correlation of the JSD and the mobility score.

- Lines 489-491

The text states that the results are recapitulated but the figure is not intuitive and the result is not well explained. A more explicit assessment of this, such as a venn diagram or bar chart displaying numbers of consistent/ discordant results might be informative.

Thank you for your advice. We added Venn diagrams alongside the original plots to show the overlap in more detail.

- Lines 510-512

This statement does not feel justified as there is not sufficient validation that the additional hits identified by RAPDOR are biologically meaningful. There would need to be a much more comprehensive comparison of the results from the two methods, as well as experimental validation of the importance of additional hits for this claim to be appropriate.

We removed this strong statement.

- Lines 521-531

It is not clear why this further dataset is included in this paper, as it is addressed in such a cursory manner. There is no overview of what the analysis with RAPDOR looked like for this data, nor is there a clear description of what the data is. The text also states that the main findings of the publication are reproduced by RAPDOR but there is only one panel in a supplementary figure of the results from this analysis and this doesn't offer any form of comparison. This only indicates replication of one of the paper's findings. If others were recapitulated, these would also need to be shown.

To focus our presentation, we removed the paragraph about this dataset and the associated previous Supplemental Fig. S6 and previous Table S5.

.....

Reviewer #2 (Remarks to the Author):

Reviewer #3 (Remarks to the Author):

A non-parametric tool for comparing fractionated mass-spec data is a very useful and exciting addition to the RNA/protein field. It may also be useful for analysing gradient-fractionated polysomal profile of RNAs and proteins between two conditions. Since the focus of this study is the development of an analysis tool, I do not have enough bioinformatic background to assess the tool at the coding level. However some questions regarding the wet-lab aspects and the data may benefit from more discussions:

1. Lysis without RNase inhibitor will cause some RNA-protein complex to dissociate during lysis. The ladder signals of individual transcripts in the Northern blot in Fig.S1 suggest they are actively being degraded, without addition of exogenous RNases.

All experimental steps were done at 4°C and as fast as possible. Although it cannot be entirely excluded that resident RNases would have been active during the first steps of the protocol, the gel analysis shown in Fig. 1B suggests that the RNA was largely intact.

2. Lysis without cycloheximide will cause ribosome run-off, leaving proteins involved in active translation out from the dataset. eg. One may expect more large subunit ribosomal proteins to appear as RNA-binding if cycloheximide was present.

Cycloheximide is blocking eukaryotic translational elongation, while here we have been working with a bacterium. But it is right that the experimental approach made no difference between actively translating ribosomes and inactive ribosomal subunits. It is totally right to assume that without stopping the translating ribosomes the putative shifting of certain factors, such as elongation and termination factors, for example, were lost.

However, even if the large subunit was not actively translating, it contains a huge RNA, the rRNA, and therefore the total digestion of this RNA should give a shift in the position of ribosomal proteins. We rather discuss that we assume that partial ribosomal subunit complexes remained because they were stabilized by protein-protein interactions, and/or because some rRNA fragments were inaccessible for the RNase treatment, consistent with some remaining RNA fragments in fraction 20 (Fig. 1C).

One sentence added (lines 341–345 in the version with tracked changes).

3. If the lysate consists of the total ribonucleoprotein in the cell, why were only 31% of the annotated proteome detected (Line 178)? Is this a loading or detection issue or is the cell stage used in this study known for expressing only 31% of the proteome?

We thank the reviewer for his detailed look at our data. Indeed, a proteome coverage of 31% seems to be lower than usually expected. However, it should be noted that *Synechocystis* 6803 was only analyzed under one condition in our experiments. Therefore, a higher proteome coverage might

have been possible if several, preferably different, conditions requiring expression of a larger part of the proteome had been considered.

Regarding the technical aspects, given the high number of samples we opted for a comparatively simple sample preparation and LC-MS instrumentation in our study. A higher coverage could possibly have been achieved by using more sophisticated methods and different types of instruments. Nevertheless, we believe that the study generated a high-quality data set that was analyzed by RAPDOR using a Jensen-Shannon Distance-based approach and helped to identify a considerable number of potential RBPs.

We have inserted one sentence to briefly refer to the first point “*Hence, ~31% of the annotated 3.681 proteins in Synechocystis 6803 were identified, which relates to the fact that only material from a single growth condition was analyzed*” (lines 188–190).

4. Presence of individual proteins/complexes in the sucrose gradient should follow a distribution spanning multiple fractions, but they appear to be single dots in Fig.S2. What do the dots represent? Mean or median fraction number? This is not clearly described in either main text or in legend.

For the calibration curve in Fig. S3 (previously Fig. S2), the respective peak fractions of the indicated proteins were selected, so the dots neither represent mean nor median fraction numbers. This is despite the fact that several of these protein complexes span also adjoining fractions.

We have slightly expanded the legend to Fig. S3 to make this clearer.

5. Not all ribosomes are actively translating (such as the case described in Line 425), so some ribosomal proteins may be at heavy fractions and only some of the same protein (from ribosomes in contact with mRNA) move to lighter fractions following RNase treatment. Is the authors' algorithm sensitive enough to pick up such case? Can this explain the lower number of ribosomal proteins classed by RAPDOR as RNA-binding compared to TripPepSVM (Line 348)?

Yes, the algorithm is sensitive enough for that. If only a fraction of an RNA-dependent protein moved to a different fraction, this would still be detected.

Having said that, one should note that the gradient technique has a resolution limit. To move to a different fraction, a difference of at least 50 kDa is required. So, if an RNP loses less than that mass, the same fraction was assigned.

The lower number of ribosomal proteins classified by RAPDOR as RNA-binding compared to TripPepSVM is rather a training effect of this algorithm. TripPepSVM generated a list of 306 candidate RBPs for *Synechocystis* 6803, among them were 45 of the 52 ribosomal proteins. The algorithm was trained based on the GO-term annotation of proteins as “RNA-binding”. Yet, it is a possibility that heterogeneity among ribosomes made some less amenable to the RNase treatment than others, consistent

with remaining RNA fragments in fraction 20 after RNase treatment (cf. Fig. 1C), and this contributed to a lowered detectability of ribosomal protein shifting, especially for those of the large subunit.

We now briefly refer to this possibility in lines 341–45.

6. RNA-binding proteins can also be detected by comparing proteins co-purified with RNA with/without UVC crosslinking (OOPS or TRAPP approach). TRAPP dataset for human HEK293T cells is available (<https://doi.org/10.1093/nar/gkad1249>). Has a RNase-gradient-based dataset been done on HEK cells? Or has UV-crosslinking approach been done on cyanobacterium? If so how do the RNA-binding proteins compare between the approaches?

Yes, another UV-crosslinking approach has been done on the cyanobacterium *Nostoc 7120* (Brenes-Álvarez et al., 2025). We now directly compare to the findings from this publication on multiple occasions. We do so in the Results section on “*Candidates for proteins interacting with RNA from GradR data*”. We also refer to the respective homologs in *Nostoc 7120* that also showed RNA-dependent shifts in Table 1 and we have added such information also in Table S3, column S, also referring to further studies.

7. Typo "her" in Line 535.

Correction done as suggested.

8. In theory the RAPDOR tool may also be able to analyse mass-spec data across a time course, instead of fractions, between two conditions. Has this been tested? The TRAPP time course dataset above may be a suitable test data.

Unfortunately, RAPDOR cannot be applied to time courses. The reason is that the sum over all fractions of a single protein should be approximately equivalent to the entire amount of that protein. This is not the case for a time course making it impossible to estimate a probability distribution over the fractions.

.....

Reviewer #4 (Remarks to the Author):

In their study, Wolfgang R. Hess and Rolf Backofen introduce RAPDOR, a novel computational analysis method tailored for proteomic datasets, particularly those stemming from gradient fractionation-based proteomics aimed at identifying RNA-binding proteins (RBPs). The core of RAPDOR lies in its utilization of the Jensen-Shannon Distance (JSD) and Analysis of Similarities (ANOSIM), augmented by additional computations such as relative entropy and Shannon entropy. As a proof-of-concept, the authors analyze data from *Synechocystis 6803*, uncovering novel RBP candidates when compared to the original GradR analysis. Furthermore, they apply RAPDOR to spatial proteomics, revisiting proteomics data from various cellular compartments, and conclude by discussing potential applications of their method, re-analyzing proteomics data from different cellular compartments. At last they discussed the potential applications of RAPDOR.

This study slightly changes the original analysis pipeline of GradR and R-Deep, mainly by introducing nonparametric analysis instead of parametric analysis. This might be of great importance in statistical theory, but its importance and novelty in biological contexts remain somewhat questionable. Besides, the outcome difference between different analysis methods is not sufficiently validated. In summary, this article would be better to be published in a journal focusing more on statistics or computation. If the article is otherwise considered for publication, the following questions and concerns must be thoroughly addressed.

We now present validation results, have more clearly worked out the comparison to other analysis workflows, and pronounce now much more the biological relevance (e.g. regarding phosphoglucomutase as a central hub in primary metabolism, two RNA modification enzymes as possible RNA chaperones, the identification of a ribosome hibernation factor and of a novel type of RNA-binding antitoxin).

Please see below for details.

Major:

1. What is the outcome difference between RAPDOR and the original analysis pipelines in GradR and R-Deep? A thorough and comprehensive analysis between different methods need to be presented in your article.

We reanalyzed the dataset published together with the R-DeeP pipeline and benchmarked both tools regarding RBP and RDP classification. In the same section we also pinpoint example scenarios where either of the tools fails to classify an RBP due to noisy proteomics measurements.

Please see the new paragraph in the results section “RAPDOR outperforms R-DeeP in RBP classification”.

2. What is your final criteria in determining whether a protein is an RBP? The selection of your cutoff should at least be mentioned and validated with regard of other established datasets.

To determine the ANOSIM cutoff, we used the 95 percentile of the distribution of ANOSIM R's from all proteins generated using every possible permutation of the treatment labels (Supplementary Fig. S5). The resulting thresholds were an ANOSIM R value >0.4815 (p-value ≤ 0.05).

To clarify our cutoff selection, we explain our method in more detail in Results and Methods, and added an entirely new segment about “*The RAPDOR ranking and selection of R-value cutoffs*” as the second section of the Discussion.

3. There are many RBP datasets generated with different methods, especially for Mus musculus and Homo Sapiens. A comparative analysis integrating your study's results, the original results, and these established datasets is necessary.

Done as suggested. Following this advice, we used the original R-DeeP dataset generated in an experiment similar to GradR in human HeLa cells for benchmarking against the R-DeeP tool.

Please see the new section in the Results “Enhanced RBP classification performance with RAPDOR”.

4. To bolster the biological significance of the findings, **biochemical experiments** should be conducted to:

- (1) Confirm that one or more of the newly identified RBPs are indeed RBPs.
- (2) Verify that one or more RBPs detected by the original analysis pipeline but not by RAPDOR are false positives.

Following this advice, we have performed biochemical experiments with eight predicted RNA-dependent proteins and performed PNK assays. The results indicated:

- (1) Direct RNA binding for five of these eight proteins.**
- (2) That one RBP detected by the original analysis pipeline (and by RAPDOR), the WD40 repeat protein SII1315, was false positive.**

For details, please see our new section “Validation of RNA binding *in vivo*“. It should be noticed that only one of these eight proteins was predicted by TripPepSVM as an RBP. Therefore, these results indicate that experiments analyzed by either R-DeeP or RAPDOR add biologically significant results compared to the prediction trained on known RBPs, but that RAPDOR had a higher sensitivity.

The fact that three of the eight proteins were not verified point at possible technical and biological effects, such as sensitivity or proteins involved in complexes with other RBPs but not binding RNA directly.

We have added a new section on “Validation of RNA binding *in vivo*“ to the Results, a corresponding section to the Materials and Methods, and adapted the text in the Discussion to refer to these new results.

Minor:

1. Please clarify the types of data that can be analyzed using RAPDOR by providing additional examples or explanations to broaden its application scope more distinctly.

We clarify these points in “The RAPDOR workflow is broadly applicable” section explaining the type of data that can be analyzed by RAPDOR. The expected input format can be found in the online documentation of the tool:

https://domonik.github.io/RAPDOR/nightly/data_preparation.html

Point-to-point replies to the reviews of our manuscript Ref: Nature Communications manuscript,

Authors: Hemm, Rabsch, et al. "RAPDOR: Using Jensen-Shannon Distance for the computational analysis of complex proteomics datasets".

REVIEWER COMMENTS

Reviewer #1 (Remarks to the Author):

Firstly, we would like to thank the authors for their thoughtful responses to our comments and commend them for the effort they have put in to addressing our concerns. The paper benefits from the improved exploration of the nature of proteins identified and from better comparison to existing datasets. However, even with the added experiments and analysis, we still feel that the strength of this paper is in introducing a new, more flexible analysis tool, rather than offering substantial biological insight. Therefore, we feel that the paper might be better suited to a more technical or bioinformatics focused journal.

We are convinced that our paper is of broad interdisciplinary interest and therefore should be published in an appropriate interdisciplinary journal.

A few specific comments that we don't feel were fully understood or addressed are also included below:

- Our suggestion of incorporating dimensionality reduction methods to understand the data holistically was meant to refer to plotting individual protein behaviour patterns, rather than just samples. This is not essential to the paper's message but might be an interesting extension in future.

Thank you for this suggestion. We would like to clarify that we have already implemented dimensionality reduction for individual proteins in the library and added a possibility to display it in our webserver. Specifically, under the "Bubble Plot" tab, users can select "PCA" as the plot type to explore protein-level variation in an interactive manner (see Figure R1 below for illustration).

Figure R1. Dimensionality reduction for individual proteins from *Synechocystis* Grad-R data based on PCA. Every dot represents an individual protein.

In our previous response, we noted that this PCA highlights peak fractions of each protein rather than fraction shifts. Because our primary interest lies in shifts, a dimensionality reduction would benefit from developing specific new tools that explicitly capture shift-related variations. While this is an interesting direction for future work, we believe 1) developing a new, shift-sensitive dimensionality reduction is out of the scope of this paper, and 2) the current Bubble Plot (see Fig. 6 in the manuscript) already provides an informative and accessible view of the shift patterns especially since the webserver version can be filtered e.g. for proteins with a high ANOSIM R.

- The addition of the PNK assay is very helpful for knowing whether this pipeline can identify new RBPs. However, it is not clear how R-Deep versus RAPDOR performs in identifying new RBPs (were the three from R-Deep those that had RNA-binding activity?). Also, as we suggested previously, this would be much more compelling if 'candidates' with very subtle shifts could be validated to indicate that these shifts are meaningful.

We experimentally verified 5, not 3 RBPs of the 8 tested candidates. All five were identified by RAPDOR, but three of these had not been predicted by the TriPepSVM algorithm. Two of the 5 experimentally validated RBPs were correctly identified by R-Deep, but R-Deep had also identified the WD40 repeat protein Sll1315, which tested negative.

Both QueF and Pgm show relatively low Jensen-Shannon distances of around 0.16—close to the median across the entire dataset—while clearly shifting proteins such as Sll1967 reach values as high as ~0.86. Thus, QueF and Pgm do qualify as candidates with very subtle shifts. Therefore, these examples underscore the sensitivity of our approach: despite only subtle distributional changes, ANOSIM successfully identifies these proteins as shifted, demonstrating the method's ability to detect biologically relevant signals even in challenging cases.

- The use of TriPepSVM over other RBP prediction tools is still not adequately justified. Perhaps it would be improved by comparing results across different tools or providing more insight into the general features of identified proteins that make them likely to be RBPs.

Yes, this comparison is available. We checked again the original TriPepSVM publication by Bressin et al. (2019), and found in Figure 2 of that publication (see Fig. R2 below) a comparison to the other existing tools like RBPPred, RNApred and SPOT-Seq-RNA. Here, TriPepSVM exhibited a superior performance, especially when looking at the Precision-Recall curves (panels D,E,F), which are more appropriate for imbalanced datasets than the ROC-curves (panels A,B,C). An AUC of 0.72 compared to AUCs below 0.43 for the competitors on *E. coli* is a large margin and great improvement in prediction quality, which led us to the decision to use TriPepSVM as tool for our purposes.

[editorial note: figure redacted]

Figure R2. Performance of TriPepSVM in comparison to other RBP prediction methods. This is Figure 2 in Bressin et al. (2019).

Moreover, unlike other bioinformatics tools (such as RBPpred or Deep-RBPpred), which use eukaryotic and prokaryotic RBPs as a training dataset, the original TriPepSVM pipeline enables users to select a specific strain's proteome and the proteomes of its closest evolutionary relatives to create a training set. This feature is crucial because RBPs in eukaryotes and prokaryotes may not rely on exactly the same features. In fact, the original TriPepSVM article (Bressin et al., 2019) showed that tripeptides enriched in RBPs are more frequently found in intrinsically disordered regions in humans, but not in *Salmonella* or *E. coli*. The ability to use a custom prokaryotic training dataset, coupled with TriPepSVM's superior performance compared to RBPpred or SPOT-seq (see Figure R2 above) are the reasons why we selected this bioinformatics tool for our analysis.

We have used a modified version of the TriPepSVM described in more detail by Brenes-Álvarez et al. (2025) using a custom training dataset. In short, instead of using a reference proteome and its closely related proteomes, we selected proteomes of cyanobacteria with diverse morphologies and lifestyles. In addition to the proteomes of our phylum of study, we also incorporated those of two extensively studied Gram-negative bacteria: *Escherichia coli* K12 and *Salmonella typhimurium* LT2. Furthermore, we optimized the specific parameters used by TriPepSVM for this training dataset using a 10-fold cross-validation setting (Brenes-Álvarez et al., 2025).

Change in manuscript: To explain these facts better, we have introduced a short additional chapter “*Bioinformatic prediction of RBPs in cyanobacteria*” in the Discussion of our revised manuscript.

Reviewer #2 (Remarks to the Author):

Reviewer #3 (Remarks to the Author):

This revision by Hemm et al addressed most of reviewers comments, and the manuscript is much better supported by experimental validations and clearer descriptions of its supported data type and limitations.

Many thanks for your comments that our revised manuscript is much better supported by experimental validations and clearer descriptions of the supported data type and limitations! This is really appreciated.

However one of my original questions was whether there is a UV-crosslinking-based dataset for RNA-interactome (eg. Oops or TRAPP), of which the stringency of RBP identification can be compared with this RNase-gradient-based method. In authors' rebuttal they cited Brenes-Álvarez et al., 2025, which did not have a UV-crosslinking-based RNA interactome, but a CLIP-Seq dataset which looked at protein-interacting RNAs, not RNA-interacting proteins, and therefore cannot be compared with the authors' dataset. A quick search online did not yield any articles applying UV-crosslinking-based methods in cyanobacterium.

Yes, the reviewer is correct with this statement. The lack of systematic analyses of RBPs and the RNA-interactome is why we felt we should look in this group of organisms.

That said, research articles using human cells can be found in abundance, including the HeLa cell UV-crosslinking-based dataset in <https://doi.org/10.1016/j.cell.2018.11.004> and the HeLa cell RNase-gradient-based dataset in <https://doi.org/10.1016/j.molcel.2019.04.018>, which was extensively cited in authors' article. It is a shame that this opportunity to compare between the two approaches is missed.

We thank the reviewer for raising this point. We would like to clarify that the RNase-gradient-based method we employ was not developed as part of our work, but rather adopted from the original publication (Gerovac et al., 2020) where its justification and validation are more detailed. While we agree that UV-crosslinking-based methods have great merit, there are also caveats in loosing information on co-fractionating proteins of the same ribonucleoprotein complex, or on the damaging doses of UV required for the pigment-rich cyanobacteria (see also our reply to the following comment below).

We also agree that systematic benchmarking of different RBP detection methods is an important and interesting topic, but believe it falls outside the scope of the current manuscript and would require a dedicated, comprehensive study on its own as highlighted in the Oops paper by Queiroz et al. (2019). Moreover, please note that the GradR method is widely in use by other groups and applied to analyses in organisms as different as yeast (Wäber et al., 2024), *E. coli* (Stenum et al., 2023), or *Enterococcus* species (Michaux et al., 2023).

We have introduced a short additional paragraph in the Discussion section to better explain these aspects.

The method described by the authors requires 40 fractions to be analysed in mass spectrometer per sample (w/wo RNase) per replicate. This is very costly compared to the UV-crosslinking-based method, with only two MS injections required (w/wo crosslink) using single-shot label free DIA. Some UV-based methods (like Oops) also selects RNA-binding proteins based on their susceptibility to RNase. Therefore unless

this RNase-gradient-based approach provides far superior sensitivity or accuracy for RBP detection, its appeal to other researchers (20x MS cost!) will be limited.

Yes, we agree that UV-crosslinking-based methods have great merit. We have been working on implementing it for cyanobacteria, but found it very challenging due to the fact that these photosynthetic bacteria are extremely rich in pigments that absorb light of various wavelengths, including UV, necessitating really high and damaging irradiation doses.

However, there is another aspect, that should not be lost here, the RNase-gradient-based approach provides information about co-fractionating proteins that could be involved in the same ribonucleoprotein complex. This is exemplified here for the ribosomal proteins and our finding of SII0947 as the cyanobacterial homolog of the ribosome hibernation factor (Hpf/RaiA/LrtA), which was co-fractionating with ribosomal proteins in fraction 19. Another candidate for a previously unknown ribosome-interacting protein is Ssr3189, which was also co-fractionating with ribosomal proteins in fraction 19 (here, a separate publication is in progress).

On the cost argument, the reviewer is right that application of DIA can reduce the necessary MS-time drastically, thereby keeping the sensitivity of peptide identification high. However, unfortunately not every proteomics lab is equipped with the necessary instruments to perform this kind of experiments. Indeed, we used an available Orbitrap Velos Pro, which is not capable of performing DIA experiments. Hence, sample fractionation is one way to still be able to sensitively detect peptides in bottom-up mass spectrometric analyses. This, of course multiplies the number of samples to be analyzed by a factor of 10. However, given the fact that each fraction of a sample is usually LC-separated with a shorter gradient compared to unfractionated samples, the net analyses time is only multiplied by a factor of 6. An alternative option to reduce the MS-time even for fractionated samples of the GradR approach would be to multiplex samples with chemical labeling techniques like TMT. This would reduce the MS-time of our experiment to 1 day as compared to ~0.5 days with the UV-crosslink-DIA experiment suggested by the reviewer, but still allow to also use MS-instrument with DDA-options only.

Change in manuscript: To explain these considerations better, we have introduced a short additional paragraph in the Discussion within the section “*RNase sensitive gradient fractionation of Synechocystis*”.

Since I did not see other reviewers writing about the app, I have done a test run in both the latest Linux/python environment and in Windows executable. In both environments the app installed and ran with no issues. The API documentation in <https://domonik.github.io/RAPDOR/> appeared to be still under construction, so I cannot comment on this part.

Thank you for testing the application in both environments and for your feedback. We would like to clarify that the API documentation is now complete and fully accessible as of release v0.1.8.

My only issue is the index webpage misses a clear link to its main GitHub page at <https://github.com/domonik/RAPDOR/>. This link is only visible at the installation page. Without this link in the index page will potentially make it more difficult for users to

report bugs or incompatibility issues, especially in the future when python or other runtime environments are updated.

Thank you for pointing this out. We've now added a clear link to the GitHub repository on the index page. Additionally, a GitHub icon in the upper right corner is present on every page and links directly to the repository for easy access.

Based on its comparison with R-Deep pipeline, I believe the RAPDOR pipeline is solid and the field will benefit from the publication of this non-parametric analysis approach for fractionated data. However without proper justification for the cost of data generation, hence the requirement for this pipeline analysing fractions, its impact in the RNA-interactome field may not be sound.

We now mention the cost aspects and other advantages/disadvantages in an added short paragraph in our Discussion.

Reviewer #3 (Remarks on code availability):

App test-ran under both Linux/python environment and in Windows executable. Installation and execution were fine. The provided test files were successfully analysed. The installation guide and tutorial were user friendly, however the API documentation is still under construction at this stage.

Thank you for testing the app thoroughly. As mentioned in our response to a similar comment from Reviwer #2 above, the API documentation has now been fully completed and made available as recommended. All functions are now documented and accessible, providing clear guidance for users.

Reviewer #4 (Remarks to the Author):

I appreciate the authors' efforts in providing additional supportive experiments and more comprehensive discussions. In this revised version, the authors have satisfactorily addressed most concerns raised. I believe this manuscript advances the methodology for RNA-dependent protein identification and validates it with wet lab experiments. Therefore, I recommend its acceptance by Nature Communications.

Many thanks for your positive and encouraging comments! Your recommendation is really appreciated.

References in this letter

- Brenes-Álvarez M, *et al.* R-DeeP/TripepSVM identifies the RNA-binding OB-fold-like protein PatR as regulator of heterocyst patterning. *Nucleic Acids Research* **53**, gkae1247 (2025).
- Bressin A, *et al.* TriPepSVM: de novo prediction of RNA-binding proteins based on short amino acid motifs. *Nucleic Acids Research* **47**, 4406–4417 (2019).
- Gerovac M., *et al.* Global discovery of bacterial RNA-binding proteins by RNase-sensitive gradient profiles reports a new FinO domain protein. *RNA* **26**, 1448–1463 (2020).
- Michaux C, *et al.* Grad-seq analysis of *Enterococcus faecalis* and *Enterococcus faecium* provides a global view of RNA and protein complexes in these two opportunistic pathogens. *microLife* 4: uqac027 (2023).

- Queiroz RML, *et al.* Comprehensive identification of RNA-protein interactions in any organism using orthogonal organic phase separation (OOPS). *Nat Biotechnol* **37**, 169–178 (2019).
- Stenum TS, *et al.* RNA interactome capture in *Escherichia coli* globally identifies RNA-binding proteins. *Nucleic Acids Res* 51: 4572–4587 (2023).
- Wäber NB, *et al.* A census of RNA-dependent proteins in yeast. *BioRxiv* 2024.12.06.627129 (2024).